# Semiparametric Differential Graph Models

**Pan Xu**
University of Virginia
px3ds@virginia.edu

**Quanquan Gu**
University of Virginia
qg5w@virginia.edu

## Abstract

In many cases of network analysis, it is more attractive to study how a network varies under different conditions than an individual static network. We propose a novel graphical model, namely Latent Differential Graph Model, where the networks under two different conditions are represented by two semiparametric elliptical distributions respectively, and the variation of these two networks (*i.e.*, differential graph) is characterized by the difference between their latent precision matrices. We propose an estimator for the differential graph based on quasi likelihood maximization with nonconvex regularization. We show that our estimator attains a faster statistical rate in parameter estimation than the state-of-the-art methods, and enjoys the oracle property under mild conditions. Thorough experiments on both synthetic and real world data support our theory.

## 1 Introduction

Network analysis has been widely used in various fields to characterize the interdependencies between a group of variables, such as molecular entities including RNAs and proteins in genetic networks [3]. Networks are often modeled as graphical models. For instance, in gene regulatory network, the gene expressions are often assumed to be jointly Gaussian. A Gaussian graphical model [18] is then employed by representing different genes as nodes and the regulation between genes as edges in the graph. In particular, two genes are conditionally independent given the others if and only if the corresponding entry of the precision matrix of the multivariate normal distribution is zero. Nevertheless, the Gaussian distribution assumption, is too restrictive in practice. For example, the gene expression values from high-throughput method, even after being normalized, do not follow a normal distribution [19, 26]. This leads to the inaccuracy in describing the dependency relationships among genes. In order to address this problem, various semiparametric Gaussian graphical models [21, 20] are proposed to relax the Gaussian distribution assumption.

On the other hand, it is well-known that the interactions in many types of networks can change under various environmental and experimental conditions [1]. Take the genetic networks for example, two genes may be positively conditionally dependent under some conditions but negatively conditionally dependent under others. Therefore, in many cases, more attention is attracted not by a particular individual network but rather by whether and how the network varies with genetic and environmental alterations [6, 15]. This gives rise to differential networking analysis, which has emerged as an important method in differential expression analysis of gene regulatory networks [9, 28].

In this paper, in order to conduct differential network analysis, we propose a Latent Differential Graph Model (LDGM), where the networks under two different conditions are represented by two transelliptical distributions [20], *i.e.*, $TE_d(\boldsymbol{\Sigma}_X^*, \xi; f_1, \ldots, f_d)$ and $TE_d(\boldsymbol{\Sigma}_Y^*, \xi; g_1, \ldots, g_d)$ respectively. Here $TE_d(\boldsymbol{\Sigma}_X^*, \xi; f_1, \ldots, f_d)$ denotes a $d$-dimensional transelliptical distribution with latent correlation matrix $\boldsymbol{\Sigma}_X^* \in \mathbb{R}^{d \times d}$, and will be defined in detail in Section 3. More specifically, the connectivity of the individual network is encoded by the latent precision matrix (*e.g.*, $\boldsymbol{\Theta}_X^* = (\boldsymbol{\Sigma}_X^*)^{-1}$) of the corresponding transelliptical distribution, such that $[\boldsymbol{\Theta}_X^*]_{jk} \neq 0$ if and only if there is an edge between the $j$-th node and the $k$-th node in the network. And the differential graph is defined as

the difference between the two latent precision matrices $\mathbf{\Delta}^* = \mathbf{\Theta}_Y^* - \mathbf{\Theta}_X^*$. Our goal is to estimate $\mathbf{\Delta}^*$ based on observations sampled from $TE_d(\mathbf{\Sigma}_X^*, \xi; f_1, \ldots, f_d)$ and $TE_d(\mathbf{\Sigma}_Y^*, \xi; g_1, \ldots, g_d)$. A simple procedure is estimating $\mathbf{\Theta}_X^*$ and $\mathbf{\Theta}_Y^*$ separately, followed by calculating their difference. However, it requires estimating $2d^2$ parameters (*i.e.*, $\mathbf{\Theta}_X^*$ and $\mathbf{\Theta}_Y^*$), while our ultimate goal is only estimating $d^2$ parameters (*i.e.*, $\mathbf{\Delta}^*$). In order to overcome this problem, we assume that the difference of the two latent precision matrices, *i.e.*, $\mathbf{\Delta}^*$ is sparse and propose to directly estimate it by quasi likelihood maximization with nonconvex penalty. The nonconvex penalty is introduced in order to correct the intrinsic estimation bias incurred by convex penalty [10, 36]. We prove that, when the true differential graph is $s$-sparse, our estimator attains $O(\sqrt{s_1/n} + \sqrt{s_2 \log d/n})$ convergence rate in terms of Frobenius norm, which is faster than the estimation error bound $O(\sqrt{s \log d/n})$ of $\ell_{1,1}$ penalty based estimator in [38]. Here $n$ is the sample size, $s_1$ is the number of entries in $\mathbf{\Delta}^*$ with large magnitude, $s_2$ is the number of entries with small magnitude and $s = s_1 + s_2$. We show that our method enjoys the oracle property under a very mild condition. Thorough numerical experiments on both synthetic and real-world data back up our theory.

The remainder of this paper is organized as follows: we review the related work in Section 2. We introduce the proposed model and the non-convex penalty in Section 3, as well as the proposed estimator. In Section 4, we present our main theories for estimation in semiparametric differential graph models. Experiments on both synthetic and real world data are provided in Section 5. Section 6 concludes with discussion.

**Notation** For $\mathbf{x} = (x_1, \ldots, x_d)^\top \in \mathbb{R}^d$ and $0 < q < \infty$, we define the $\ell_0$, $\ell_q$ and $\ell_\infty$ vector norms as $\|\mathbf{x}\|_0 = \sum_{i=1}^d \mathbb{1}(x_i \neq 0), \|\mathbf{x}\|_q = \left(\sum_{i=1}^d |x_i|^q\right)^{1/q}$, and $\|\mathbf{x}\|_\infty = \max_{1 \leq i \leq d} |x_i|$, where $\mathbb{1}(\cdot)$ is the indicator function. For $\mathbf{A} = (A_{ij}) \in \mathbb{R}^{d \times d}$, we define the matrix $\ell_{0,0}$, $\ell_{1,1}$, $\ell_{\infty,\infty}$ and $\ell_F$ norms as: $\|\mathbf{A}\|_{0,0} = \sum_{i,j=1}^d \mathbb{1}(A_{ij} \neq 0)$, $\|\mathbf{A}\|_{1,1} = \sum_{i,j=1}^d |A_{ij}|$, $\|\mathbf{A}\|_{\infty,\infty} = \max_{1 \leq i,j \leq d} |A_{ij}|$, and $\|\mathbf{A}\|_F = \sqrt{\sum_{ij} |A_{ij}|^2}$. The induced norm for matrix is defined as $\|\mathbf{A}\|_q = \max_{\|\mathbf{x}\|_q = 1} \|\mathbf{A}\mathbf{x}\|_q$, for $0 < q < \infty$. For a set of tuples $S$, $\mathbf{A}_S$ denotes the set of numbers $[A_{(jk)}]_{(jk) \in S}$, and $\text{vec}(S)$ is the vectorized index set of $S$.

## 2 Related Work

There exist several lines of research for differential network analysis. One natural procedure is to estimate the two networks (i.e., two precision matrices) respectively by existing estimators such as graphical Lasso [12] and node-wise regression [25]. Another family of methods jointly estimates the two networks by assuming that they share common structural patterns and therefore uses joint likelihood maximization with group lasso penalty or group bridge penalty [7, 8, 14]. Based on the estimated precision matrices, the differential graph can be obtained by calculating their difference. However, both of these two types of methods suffer from the drawback that they need to estimate twice the number of parameters, and hence require roughly doubled observations to ensure the estimation accuracy. In order to address this drawback, some methods are proposed to estimate the difference of matrices directly [38, 35, 22, 11]. For example, [38] proposed a Dantzig selector type estimator for estimating the difference of the precision matrices directly. [35] proposed a D-Trace loss [37] based estimator for the difference of the precision matrices. Compared with [38, 35], our estimator is advantageous in the following aspects: (1) our model relaxes the Gaussian assumption by representing each network as a transelliptical distribution, while [38, 35] are restricted to Gaussian distribution. Thus, our model is more general and robust; and (2) by employing nonconvex penalty, our estimator achieves a sharper statistical rate than theirs. Rather than the Gaussian graphical model or its semiparametric extension, [22, 11] studied the estimation of change in the dependency structure between two high dimensional Ising models.

## 3 Semiparametric Differential Graph Models

In this section, we will first review the transelliptical distribution and present our semiparametric differential graph model. Then we will present the estimator for differential graph, followed by the introduction to nonconvex penalty.

### 3.1 Transelliptical Distribution

To briefly review the transelliptical distribution, we begin with the definition of elliptical distribution.

**Definition 3.1** (Elliptical distribution). Let $\boldsymbol{\mu} \in \mathbb{R}^d$ and $\boldsymbol{\Sigma}^* \in \mathbb{R}^{d \times d}$ with $\text{rank}(\boldsymbol{\Sigma}^*) = q \leq d$. A random vector $\boldsymbol{X} \in \mathbb{R}^d$ follows an elliptical distribution, denoted by $EC_d(\boldsymbol{\mu}, \boldsymbol{\Sigma}^*, \xi)$, if it can be represented as $\boldsymbol{X} = \boldsymbol{\mu} + \xi \mathbf{A} \mathbf{U}$, where $\mathbf{A}$ is a deterministic matrix satisfying $\mathbf{A}^\top \mathbf{A} = \boldsymbol{\Sigma}^*$, $\mathbf{U}$ is a random vector uniformly distributed on the unit sphere in $\mathbb{R}^q$, and $\xi \perp \mathbf{U}$ is a random variable.

Motivated by the extension from Gaussian distribution to nonparanormal distribution [21], [20] proposed a semiparametric extension of elliptical distribution, which is called transelliptical distribution.

**Definition 3.2** (Transelliptical distribution). A random vector $\boldsymbol{X} = (X_1, X_2, \ldots, X_d)^\top \in \mathbb{R}^d$ is transelliptical, denoted by $TE_d(\boldsymbol{\Sigma}^*, \xi; f_1, \ldots, f_d)$, if there exists a set of monotone univariate functions $f_1, \ldots, f_d$ and a nonnegative random variable $\xi$, such that $(f_1(X_1), \ldots, f_d(X_d))^\top$ follows an elliptical distribution $EC_d(\mathbf{0}, \boldsymbol{\Sigma}^*, \xi)$.

## 3.2 Kendall's tau Statistic

In semiparametric setting, the Pearson's sample covariance matrix can be inconsistent in estimating $\boldsymbol{\Sigma}^*$. Given $n$ independent observations $\boldsymbol{X}_1, ..., \boldsymbol{X}_n$, where $\boldsymbol{X}_i = (X_{i1}, ..., X_{id})^\top \sim TE_d(\boldsymbol{\Sigma}^*, \xi; f_1, \ldots, f_d)$, [20] proposed a rank-based estimator, the Kendall's tau statistic, to estimate $\boldsymbol{\Sigma}^*$, due to its invariance under monotonic marginal transformations. The Kendall's tau estimator is defined as

$$\widehat{\tau}_{jk} = \frac{2}{n(n-1)} \sum_{1 \leq i < i' \leq n} \text{sign}\left[ (X_{ij} - X_{i'j})(X_{ik} - X_{i'k}) \right]. \tag{3.1}$$

It has been shown that $\widehat{\tau}_{jk}$ is an unbiased estimator of $\tau_{jk} = 2/\pi \arcsin(\Sigma^*_{jk})$ [20], and the correlation matrix $\boldsymbol{\Sigma}^*$ can be estimated by $\widehat{\boldsymbol{\Sigma}} = [\widehat{\Sigma}_{jk}] \in \mathbb{R}^{d \times d}$, where

$$\widehat{\Sigma}_{jk} = \sin\left( \frac{\pi}{2} \widehat{\tau}_{jk} \right). \tag{3.2}$$

We use $\mathbf{T}^*$ to denote the matrix with entries $\tau_{jk}$ and $\widehat{\mathbf{T}}$ with entries $\widehat{\tau}_{jk}$, for $j, k = 1, \ldots d$.

## 3.3 Latent Differential Graph Models and the Estimator

Now we are ready to formulate our differential graph model. Assume that $d$ dimensional random vectors $\boldsymbol{X}$ and $\boldsymbol{Y}$ satisfy $\boldsymbol{X} \sim TE_d(\boldsymbol{\Sigma}^*_X, \xi; f_1, \ldots, f_d)$ and $\boldsymbol{Y} \sim TE_d(\boldsymbol{\Sigma}^*_Y, \xi; g_1, \ldots, g_d)$. The differential graph is defined to be the difference of the two latent precision matrices,

$$\boldsymbol{\Delta}^* = \boldsymbol{\Theta}^*_Y - \boldsymbol{\Theta}^*_X, \tag{3.3}$$

where $\boldsymbol{\Theta}^*_X = \boldsymbol{\Sigma}^{*-1}_X$ and $\boldsymbol{\Theta}^*_Y = \boldsymbol{\Sigma}^{*-1}_Y$. It immediately implies

$$\boldsymbol{\Sigma}^*_X \boldsymbol{\Delta}^* \boldsymbol{\Sigma}^*_Y - (\boldsymbol{\Sigma}^*_X - \boldsymbol{\Sigma}^*_Y) = \mathbf{0}, \text{ and } \boldsymbol{\Sigma}^*_Y \boldsymbol{\Delta}^* \boldsymbol{\Sigma}^*_X - (\boldsymbol{\Sigma}^*_X - \boldsymbol{\Sigma}^*_Y) = \mathbf{0}. \tag{3.4}$$

Given i.i.d. copies $\boldsymbol{X}_1, \ldots, \boldsymbol{X}_{n_X}$ of $\boldsymbol{X}$, and i.i.d. copies $\boldsymbol{Y}_1, \ldots, \boldsymbol{Y}_{n_Y}$ of $\boldsymbol{Y}$, without loss of generality, we assume $n_X = n_Y = n$, and we denote the Kendall's tau correlation matrices defined in (3.2) as $\widehat{\boldsymbol{\Sigma}}_X$ and $\widehat{\boldsymbol{\Sigma}}_Y$. Following (3.4), a reasonable procedure for estimating $\boldsymbol{\Delta}^*$ is to solve the following equation for $\boldsymbol{\Delta}$

$$\frac{1}{2}\widehat{\boldsymbol{\Sigma}}_X \boldsymbol{\Delta} \widehat{\boldsymbol{\Sigma}}_Y + \frac{1}{2}\widehat{\boldsymbol{\Sigma}}_Y \boldsymbol{\Delta} \widehat{\boldsymbol{\Sigma}}_X - (\widehat{\boldsymbol{\Sigma}}_X - \widehat{\boldsymbol{\Sigma}}_Y) = \mathbf{0}, \tag{3.5}$$

where we add up the two equations in (3.4) and replace the latent population correlation matrices $\boldsymbol{\Sigma}^*_X, \boldsymbol{\Sigma}^*_Y$ with the Kendall's tau estimators $\widehat{\boldsymbol{\Sigma}}_X, \widehat{\boldsymbol{\Sigma}}_Y$. Note that (3.5) is a Z-estimator [30], which can be translated into a M-estimator, by noticing that $1/2\widehat{\boldsymbol{\Sigma}}_X \boldsymbol{\Delta} \widehat{\boldsymbol{\Sigma}}_Y + 1/2\widehat{\boldsymbol{\Sigma}}_Y \boldsymbol{\Delta} \widehat{\boldsymbol{\Sigma}}_X - (\widehat{\boldsymbol{\Sigma}}_X - \widehat{\boldsymbol{\Sigma}}_Y)$ can be seen as a score function of the following quasi log likelihood function

$$\ell(\boldsymbol{\Delta}) = \frac{1}{2} \text{tr}(\boldsymbol{\Delta} \widehat{\boldsymbol{\Sigma}}_Y \boldsymbol{\Delta} \widehat{\boldsymbol{\Sigma}}_X) - \text{tr}\left( \boldsymbol{\Delta}(\widehat{\boldsymbol{\Sigma}}_X - \widehat{\boldsymbol{\Sigma}}_Y) \right). \tag{3.6}$$

Let $S = \text{supp}(\boldsymbol{\Delta}^*)$, in this paper, we assume that $\boldsymbol{\Delta}^*$ is sparse, *i.e.*, $|S| \leq s$ with $s > 0$. Based on (3.6), we propose to estimate $\boldsymbol{\Delta}^*$ by the following M-estimator with non-convex penalty

$$\widehat{\boldsymbol{\Delta}} = \operatorname*{argmin}_{\boldsymbol{\Delta} \in \mathbb{R}^{d \times d}} \frac{1}{2} \text{tr}(\boldsymbol{\Delta} \widehat{\boldsymbol{\Sigma}}_Y \boldsymbol{\Delta} \widehat{\boldsymbol{\Sigma}}_X) - \text{tr}\left( \boldsymbol{\Delta}(\widehat{\boldsymbol{\Sigma}}_X - \widehat{\boldsymbol{\Sigma}}_Y) \right) + \mathcal{G}_\lambda(\boldsymbol{\Delta}), \tag{3.7}$$

where $\lambda > 0$ is a regularization parameter and $\mathcal{G}_\lambda$ is a decomposable nonconvex penalty function, i.e., $\mathcal{G}_\lambda(\boldsymbol{\Delta}) = \sum_{j,k=1}^d g_\lambda(\Delta_{jk})$, such as smoothly clipped absolute deviation (SCAD) penalty [10] or minimax concave penalty (MCP) [36]. The key property of the nonconvex penalty is that it can avoid over-penalization when the magnitude is very large. It has been shown in [10, 36, 33] that the nonconvex penalty is able to alleviate the estimation bias and attain a refined statistical rate of convergence. The nonconvex penalty $g_\lambda(\delta)$ can be further decomposed as the sum of the $\ell_1$ penalty and a concave component $h_\lambda(\delta)$, i.e., $g_\lambda(\delta) = \lambda|\delta| + h_\lambda(\delta)$. Take MCP penalty for example. The corresponding $g_\lambda(\delta)$ and $h_\lambda(\delta)$ are defined as follows

$$g_\lambda(\delta) = \lambda \int_0^{|\delta|} \left( 1 - \frac{z}{\lambda b} \right)_+ dz, \text{ for any } \delta \in \mathbb{R},$$

where $\lambda > 0$ is the regularization parameter and $b > 0$ is a fixed parameter, and

$$h_\lambda(\delta) = -\frac{\delta^2}{2b} \mathbb{1}(|\delta| \le b\lambda) + \left( \frac{b\lambda^2}{2} - \lambda|\delta| \right) \mathbb{1}(|\delta| > b\lambda).$$

In Section 4, we will show that the above family of nonconvex penalties satisfies certain common regularity conditions on $g_\lambda(\beta)$ as well as its concave component $h_\lambda(\beta)$.

We will show in the next section that when the parameters of the nonconvex penalty are appropriately chosen, (3.7) is an unconstrained convex optimization problem. Thus it can be solved by the proximal gradient descent [4] very efficiently. In addition, it is easy to check that the estimator $\widehat{\boldsymbol{\Delta}}$ from (3.7) is symmetric. So it does not need the symmetrizing process adopted in [38], which can undermine the estimation accuracy.

## 4 Main Theory

In this section, we present our main theories. Let $S = \text{supp}(\boldsymbol{\Delta}^*)$ be the support of the true differential graph. We introduce the following oracle estimator of $\boldsymbol{\Delta}^*$:

$$\widehat{\boldsymbol{\Delta}}_O = \underset{\text{supp}(\boldsymbol{\Delta}) \subseteq S}{\text{argmin}} \ \ell(\boldsymbol{\Delta}), \tag{4.1}$$

where $\ell(\boldsymbol{\Delta}) = 1/2 \, \text{tr}(\boldsymbol{\Delta} \widehat{\boldsymbol{\Sigma}}_Y \boldsymbol{\Delta} \widehat{\boldsymbol{\Sigma}}_X) - \text{tr}\big(\boldsymbol{\Delta}(\widehat{\boldsymbol{\Sigma}}_X - \widehat{\boldsymbol{\Sigma}}_Y)\big)$. The oracle estimator $\widehat{\boldsymbol{\Delta}}_O$ is not a practical estimator, since we do not know the true support in practice. An estimator is said to have the oracle property, if it is identical to the oracle estimator $\widehat{\boldsymbol{\Delta}}_O$ under certain conditions. We will show that our estimator enjoys the oracle property under a mild condition.

We first lay out some assumptions that are required through our analysis.

**Assumption 4.1.** There exist constants $\kappa_1, \kappa_2 > 0$ such that $\kappa_1 \le \lambda_{\min}(\boldsymbol{\Sigma}_X^*) \le \lambda_{\max}(\boldsymbol{\Sigma}_X^*) \le 1/\kappa_1$ and $\kappa_2 \le \lambda_{\min}(\boldsymbol{\Sigma}_Y^*) \le \lambda_{\max}(\boldsymbol{\Sigma}_Y^*) \le 1/\kappa_2$. The true covariance matrices have bounded $\ell_1$ norm, i.e., $\|\boldsymbol{\Sigma}_X^*\|_1 \le \sigma_X$, $\|\boldsymbol{\Sigma}_Y^*\|_1 \le \sigma_Y$, where $\sigma_X, \sigma_Y > 0$ are constants. And the true precision matrices have bounded matrix $\ell_1$-norm, i.e., $\|\boldsymbol{\Theta}_X^*\|_1 \le \theta_X$ and $\|\boldsymbol{\Theta}_Y^*\|_1 \le \theta_Y$, where $\theta_X, \theta_Y > 0$ are constants.

The first part of Assumption 4.1 requires that the smallest eigenvalues of the correlation $\boldsymbol{\Sigma}_X^*, \boldsymbol{\Sigma}_Y^*$ are bounded below from zero, and their largest eigenvalues are finite. This assumptions is commonly imposed in the literature for the analysis of graphical models [21, 27].

**Assumption 4.2.** The true difference matrix $\boldsymbol{\Delta}^* = \boldsymbol{\Sigma}_Y^{*-1} - \boldsymbol{\Sigma}_X^{*-1}$ has $s$ nonzero entries, i.e., $\|\boldsymbol{\Delta}^*\|_{0,0} \le s$ and has bounded $\ell_{1,1}$ norm, i.e., $\|\boldsymbol{\Delta}^*\|_{1,1} \le M$, where $M > 0$ does not depend on $d$.

Assumption 4.2 requires the differential graph to be sparse. This is reasonable in differential network analysis where the networks only vary slightly under different conditions.

The next assumption is about regularity conditions on the nonconvex penalty $g_\lambda(\delta)$. Recall that $g_\lambda(\delta)$ can be written as $g_\lambda(\delta) = \lambda|\delta| + h_\lambda(\delta)$.

**Assumption 4.3.** $g_\lambda(\delta)$ and its concave component $h_\lambda(\delta)$ satisfy:

  (a) There exists a constant $\nu$ such that $g_\lambda'(\delta) = 0$, for $|\delta| \ge \nu > 0$.

  (b) There exists a constant $\zeta_- \ge 0$ such that $h_\lambda(\delta) + \zeta_-/2 \cdot \delta^2$ is convex.

(c) $h_\lambda(\delta)$ and $h'_\lambda(\delta)$ pass through the origin, i.e., $h_\lambda(0) = h'_\lambda(0) = 0$.

(d) $h'_\lambda(\delta)$ is bounded, i.e., $|h'_\lambda(\delta)| \leq \lambda$ for any $\delta$.

Similar assumptions have been made in [23, 33]. Note that condition (b) in Assumption 4.3 is weaker than the smoothness condition in [33], since here it does not require $h_\lambda(\delta)$ to be twice differentiable. Assumption 4.3 holds for a variety of nonconvex penalty functions including MCP and SCAD. In particular, MCP penalty satisfies Assumption 4.3 with $\nu = b\lambda$ and $\zeta_- = 1/b$. Furthermore, according to condition (b), if $\zeta_-$ is smaller than the modulus of the restricted strong convexity for $\ell(\Delta)$, (3.7) will become a convex optimization problem, even though $\mathcal{G}_\lambda(\Delta)$ is nonconvex. Take MCP for example, this can be achieved by choosing a sufficiently large $b$ in MCP such that $\zeta_-$ is small enough.

Now we are ready to present our main theories. We first show that under a large magnitude condition on nonzero entries of the true differential graph $\Delta^*$, our estimator attains a faster convergence rate, which matches the minimax rate in the classical regime.

**Theorem 4.4.** Suppose Assumptions 4.1 and 4.2 hold, and the nonconvex penalty $\mathcal{G}_\lambda(\Delta)$ satisfies conditions in Assumption 4.3. If nonzero entries of $\Delta^*$ satisfy $\min_{(j,k) \in S} |\Delta^*_{jk}| \geq \nu + C\theta_X^2 \theta_Y^2 \sigma_X \sigma_Y M \sqrt{\log s/n}$, for the estimator $\widehat{\Delta}$ in (3.7) with the regularization parameter satisfying $\lambda = 2CM\sqrt{\log d/n}$ and $\zeta_- \leq \kappa_1 \kappa_2/2$, we have that

$$\|\widehat{\Delta} - \Delta^*\|_{\infty,\infty} \leq 2\sqrt{10}\pi \theta_X^2 \theta_Y^2 \sigma_X \sigma_Y M \sqrt{\frac{\log s}{n}}$$

holds with probability at least $1 - 2/s$. Furthermore, we have that

$$\|\widehat{\Delta} - \Delta^*\|_F \leq \frac{C_1 M}{\kappa_1 \kappa_2} \sqrt{\frac{s}{n}}$$

holds with probability at least $1 - 3/s$, where $C_1$ is an absolute constant.

**Remark 4.5.** Theorem 4.4 suggests that under the large magnitude assumption, the statistical rate of our estimator is $O(\sqrt{s/n})$ in terms of Frobenius norm. This is faster than the rate $O(\sqrt{s \log d/n})$ in [38] which matches the minimax lower bound for sparse differential graph estimation. Note that our faster rate is not contradictory to the minimax lower bound, because we restrict ourselves to a smaller class of differential graphs, where the magnitude of the nonzero entries is sufficiently large.

We further show that our estimator achieves oracle property under mild conditions.

**Theorem 4.6.** Under the same conditions of Theorem 4.4, for the estimator $\widehat{\Delta}$ in (3.7) and the oracle estimator $\widehat{\Delta}_O$ in (4.1), we have with probability at least $1 - 3/s$ that $\widehat{\Delta} = \widehat{\Delta}_O$, which further implies $\text{supp}(\widehat{\Delta}) = \text{supp}(\widehat{\Delta}_O) = \text{supp}(\Delta^*)$.

Theorem 4.6 suggests that our estimator is identical to the oracle estimator in (4.1) with high probability, when the nonzero entries in $\Delta^*$ satisfy $\min_{(j,k) \in S} |\Delta^*_{jk}| \geq \nu + C\theta_X^2 \theta_Y^2 \sigma_X \sigma_Y M \sqrt{\log s/n}$. This condition is optimal up to the logarithmic factor $\sqrt{\log s}$.

Now we turn to the general case when the nonzero entries of $\Delta^*$ have both large and small magnitudes. Define $S^c = \{(j,k) : j, k = 1, \ldots, d\} \setminus S$, $S_1 = \{(j,k) \in S : |\Delta^*_{jk}| > \nu\}$, and $S_2 = \{(j,k) \in S : |\Delta^*_{jk}| \leq \nu\}$. Denote $|S_1| = s_1$ and $|S_2| = s_2$. Clearly, we have $s = s_1 + s_2$.

**Theorem 4.7.** Suppose Assumptions 4.1 and 4.2 hold, and the nonconvex penalty $\mathcal{G}_\lambda(\Delta)$ satisfies conditions in Assumption 4.3. For the estimator in (3.7) with the regularization parameter $\lambda = 2CM\sqrt{\log d/n}$ and $\zeta_- \leq \kappa_1 \kappa_2/4$, we have that

$$\|\widehat{\Delta} - \Delta^*\|_F \leq \frac{16\sqrt{3}\pi M}{\kappa_1 \kappa_2} \sqrt{\frac{s_1}{n}} + \frac{10\pi MC}{\kappa_1 \kappa_2} \sqrt{\frac{s_2 \log d}{n}}$$

holds with probability at least $1 - 3/s_1$, where $C$ is an absolute constant.

**Remark 4.8.** Theorem 4.7 indicates that when the large magnitude condition does not hold, our estimator is still able to attain a faster rate. Specifically, for those nonzero entries of $\Delta^*$ with large magnitude, the estimation error bound in terms of Frobenius norm is $O(\sqrt{s_1/n})$, which is the same

as the bound in Theorem 4.4. For those nonzero entries of $\boldsymbol{\Delta}^*$ with small magnitude, the estimation error is $O(\sqrt{s_2 \log d/n})$, which matches the convergence rate in [38]. Overall, our estimator obtains a refined rate of convergence rate $O(\sqrt{s_1/n} + \sqrt{s_2 \log d/n})$, which is faster than [38]. In particular, if $s_2^* = 0$, the refined convergence rate in Theorem 4.7 reduces to the faster rate in Theorem 4.4.

## 5 Experiments

In this section, we test our method on both synthetic and real world data. We conducted experiments for our estimator using both SCAD and MCP penalties. We did not find any significant difference in the results and thus we only report the results of our estimator with MCP penalty. To choose the tuning parameters $\lambda$ and $b$, we adopt 5-fold cross-validation. Denoting our estimator with MCP penalty by **LDGM-MCP**, we compare it with the following methods: (1) **SepGlasso**: estimating the latent precision matrices separately using graphical Lasso and Kendall's tau correlation matrices [20], followed by calculating their difference; (2) **DPM**: directly estimating differential precision matrix [38]. In addition, we also test differential graph model with $\ell_{1,1}$ penalty, denoted as **LDGM-L1**. Note that LDGM-L1 is a special case of our method, since $\ell_{1,1}$ norm penalty is a special case of MCP penalty when $b = \infty$. The LDGM-MCP and LDGM-L1 estimators are obtained by solving the proximal gradient descent algorithm [4]. The implementation of DPM estimator is obtained from the author's website, and the SepGlasso estimator is implemented by graphical Lasso.

### 5.1 Simulations

We first show the results on synthetic data. Since the transelliptical distribution includes Gaussian distribution, it is natural to show that our approach also works well for the latter one. We consider the dimension settings $n = 100$, $d = 100$ and $n = 200$, $d = 400$ respectively. Specifically, data are generated as follows: (1) For the Gaussian distribution, we generate data $\{\boldsymbol{X}_i\}_{i=1}^n \sim N(\boldsymbol{0}, \boldsymbol{\Sigma}_X^*)$ and $\{\boldsymbol{Y}_i\}_{i=1}^n \sim N(\boldsymbol{0}, \boldsymbol{\Sigma}_Y^*)$ with precision matrices $\boldsymbol{\Sigma}_X^{*-1}$ and $\boldsymbol{\Sigma}_Y^{*-1}$ generated by **huge** package [1]. (2) For the transelliptical distribution, we consider the following generating scheme: $\{\boldsymbol{X}_i\}_{i=1}^n \sim TE_d(\boldsymbol{\Sigma}_X^*, \xi; f_1, \ldots, f_d)$, $\{\boldsymbol{Y}_i\}_{i=1}^n \sim TE_d(\boldsymbol{\Sigma}_Y^*, \xi; g_1, \ldots, g_d)$, where $\xi \sim \chi_d$, $f_1^{-1}(\cdot) = \ldots = f_d^{-1} = \text{sign}(\cdot)|\cdot|^3$ and $g_1^{-1}(\cdot) = \ldots = g_d^{-1}(\cdot) = \text{sign}(\cdot)|\cdot|^{1/2}$. The latent precision matrices $\boldsymbol{\Sigma}_X^{*-1}$ and $\boldsymbol{\Sigma}_Y^{*-1}$ are generated in the same way as the Gaussian data. For both Gaussian and transelliptical differential graph mdoels, we consider two settings for individual graph structures: (1) both $\boldsymbol{\Sigma}_X^{*-1}$ and $\boldsymbol{\Sigma}_Y^{*-1}$ have "random" structures; (2) $\boldsymbol{\Sigma}_X^{*-1}$ has a "band" structure, $\boldsymbol{\Sigma}_Y^{*-1}$ has a "random" structure.

Given an estimator $\widehat{\boldsymbol{\Delta}}$, we define the true positive and negative rates of $\widehat{\boldsymbol{\Delta}}$ as

$$\text{TP} = \frac{\sum_{j,k=1}^d \mathbb{1}(\widehat{\Delta}_{jk} \neq 0 \text{ and } \Delta_{jk}^* \neq 0)}{\sum_{j,k=1}^d \mathbb{1}(\Delta_{jk}^* \neq 0)}, \qquad \text{TN} = \frac{\sum_{j,k=1}^d \mathbb{1}(\widehat{\Delta}_{jk} = 0 \text{ and } \Delta_{jk}^* = 0)}{\sum_{j,k=1}^d \mathbb{1}(\Delta_{jk}^* = 0)}.$$

The receiver operating characteristic (ROC) curves for transelliptical differential graph models are shown in Figure 1, which report the performances of different methods on support recovery. The ROC curves were plotted by averaging the results over 10 repetitions. From Figure 1 we can see our estimator (LDGM-MCP) outperforms other methods in all settings. In addition, LDGM-L1 as a special case of our estimator also performs better than DPM and SepGlasso, although it is inferior to LDGM-MCP because the MCP penalty can correct the bias in the estimation and achieve faster rate of convergence. Note that SepGlasso's performace is poor since it highly depends on the sparsity of both individual graphs. When $n > 100$, the DPM method failed to output the solution in one day and thus no result was presented. This computational burden is also stated in their paper. We use the Frobenius norm $\|\widehat{\boldsymbol{\Delta}} - \boldsymbol{\Delta}^*\|_F$ and infinity norm $\|\widehat{\boldsymbol{\Delta}} - \boldsymbol{\Delta}^*\|_{\infty,\infty}$ of estimation errors to evaluate the performances of different methods in estimation. The results averaged over 10 replicates for transelliptical differential graph are summarized in Tables 1 and 2 respectively. Our estimator also achieves smaller error than the other baselines in all settings. Due to the space limit, we defer the experiment results for Gaussian differential graph model to the appendix.

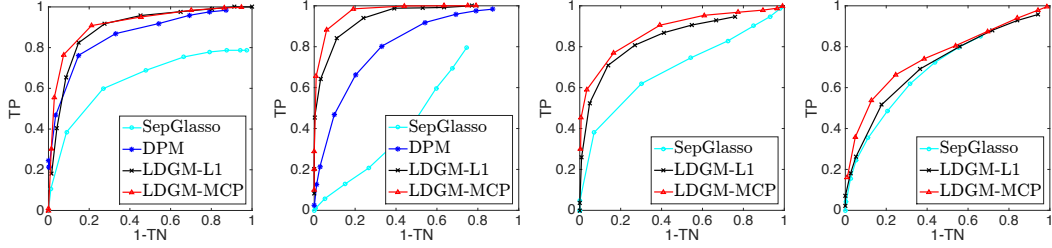

(a) Setting 1: n=100,d=100 (b) Setting 2: n=100,d=100 (c) Setting 1: n=200,d=400 (d) Setting 2:n=200,d=400

Figure 1: ROC curves for transelliptical differential graph models of all the 4 methods. There are two settings of graph structure. Note that DPM is not scalable to $d = 400$.

Table 1: Comparisons of estimation errors in Frobenius norm $\|\widehat{\boldsymbol{\Delta}} - \boldsymbol{\Delta}^*\|_F$ for transelliptical differential graph models. N/A means the algorithm did not output the solution in one day.

| | $n = 100, d = 100$ | | $n = 200, d = 400$ | |
|---|---|---|---|---|
| Methods | Setting 1 | Setting 2 | Setting 1 | Setting 2 |
| SepGlasso | 13.5730±0.6376 | 25.6664±0.6967 | 22.1760±0.3839 | 39.9847±0.1856 |
| DPM | 12.7219±0.3704 | 23.0548±0.2669 | N/A | N/A |
| LDGM-L1 | 12.0738±0.4955 | 22.3748±0.6643 | 20.6537±0.3778 | 31.7630±0.0715 |
| LDGM-MCP | 11.2831±0.3919 | 19.6154±0.5106 | 20.1071±0.4303 | 28.8676±0.1425 |

Table 2: Comparisons of estimation errors in infinity norm $\|\widehat{\boldsymbol{\Delta}} - \boldsymbol{\Delta}^*\|_{\infty,\infty}$ for transelliptical differential graph models. N/A means the algorithm did not output the solution in one day.

| | $n = 100, d = 100$ | | $n = 200, d = 400$ | |
|---|---|---|---|---|
| Methods | Setting 1 | Setting 2 | Setting 1 | Setting 2 |
| SepGlasso | 2.7483±0.0575 | 8.0522±0.1423 | 2.1409±0.0906 | 6.0108±0.1925 |
| DPM | 2.3138±0.0681 | 6.3250±0.0560 | N/A | N/A |
| LDGM-L1 | 2.2193±0.0850 | 6.0716±0.1150 | 1.8876±0.0907 | 5.1858±0.0218 |
| LDGM-MCP | 1.7010±0.0149 | 4.6522±0.1337 | 1.7339±0.0061 | 4.0133±0.0521 |

## 5.2 Experiments on Real World Data

We applied our approach to the same gene expression data used in [38], which were collected from patients with stage III or IV ovarian cancer. [29] identified six molecular subtypes of ovarian cancer in this data, labeled C1 through C6. In particular, the C1 subtype was found to have much shorter survival times, and was characterized by differential expression of genes associated with stromal and immune cell types. In this experiment, we intended to investigate whether the C1 subtype was also associated with the genetic differential networks. The subjects were divided into two groups: Group 1 with $n_1 = 78$ patients containing C1 subtype, and Group 2 with $n_2 = 113$ patients containing C2 through C6 subtypes. We analyzed two pathways from the KEGG pathway database [16, 17] respectively. In each pathway, we applied different methods to determine whether there is any difference in the conditional dependency relationships of the gene expression levels between the aforementioned Group 1 and Group 2. Two genes were connected in the differential network if their conditional dependency relationship given the others changed in either magnitude or sign. In order to obtain a clear view of the differential graph, we only plotted genes whose conditional dependency with others changed between the two groups. To interpret the results, the genes associated with more edges in the differential networks were considered to be more important.

Figure 2 shows the results of estimation for the differential graph of the TGF-$\beta$ pathway, where the number of genes $d = 80$ is greater than $n_1$, the sample size of Group 1. LDGM-MCP identified two important genes, COMP and THBS2, both of which have been suggested to be related to resistance to platinum-based chemotherapy in epithelial ovarian cancer by [24]. LDGM-L1 suggested that COMP

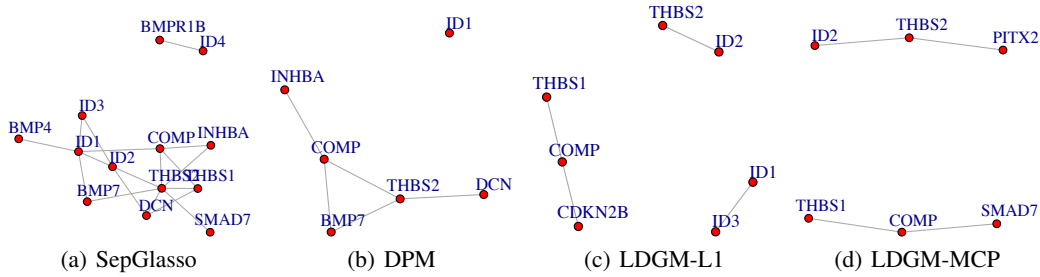

(a) SepGlasso        (b) DPM        (c) LDGM-L1        (d) LDGM-MCP

Figure 2: Estimates of the differential networks between Group 1 and Group 2. Dataset: KEGG 04350, TGF-$\beta$ pathway.

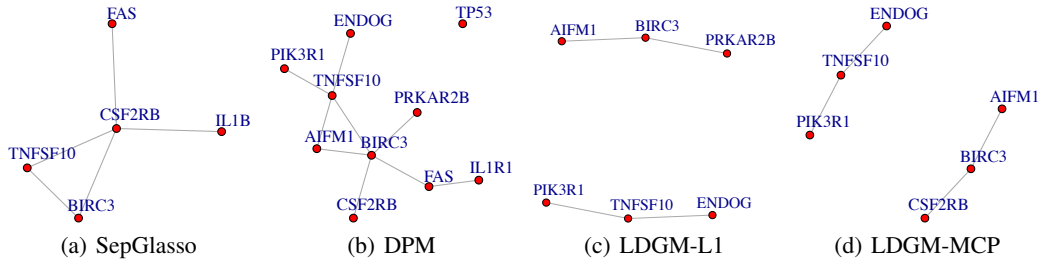

(a) SepGlasso        (b) DPM        (c) LDGM-L1        (d) LDGM-MCP

Figure 3: Estimates of the differential networks between Group 1 and Group 2. Dataset: KEGG 04210, Apoptosis pathway.

was important, and DPM also suggested COMP and THBS2. Separate estimation (SepGlasso) gave a relatively dense network, which made it hard to say which genes are more important.

Figure 3 shows the results for the Apoptosis pathway, where the number of genes $d = 87$ is also greater than $n_1$. LDGM-MCP indicated that TNFSF10 and BIRC3 were the most important. Indeed, both TNFSF10 and BRIC3 have been widely studied for use as a therapeutic target in cancer [5, 32]. LDGM-L1 and DPM also suggested TNFSF10 and BRIC3 were important. The results of LDGM-MCP, LDGM-L1 and DPM are comparable. In order to overcome the nonsparsity issue encountered in TGF-$\beta$ experiment, the SepGlasso estimator was thresholded more than the other methods. However, it still performed poorly and identified the wrong gene CSF2RB.

# 6 Conclusions

In this paper, we propose a semiparametric differential graph model and an estimator for the differential graph based on quasi likelihood maximization. We employ a nonconvex penalty in our estimator, which results in a faster rate for parameter estimation than existing methods. We also prove that the proposed estimator achieves oracle property under a mild condition. Experiments on both synthetic and real world data further support our theory.

**Acknowledgments** We would like to thank the anonymous reviewers for their helpful comments. Research was supported by NSF grant III-1618948.

## Footnotes

[1] Available on http://cran.r-project.org/web/packages/huge

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
