[Supplementary Material · Differ_Graph_supply.pdf]

# A Proof of Main Theorems

In this section, we provide the proof for our main theories.

We start by defining some notations. Note that the estimator in (3.7) can be rewritten as

$$\widehat{\boldsymbol{\Delta}} = \underset{\boldsymbol{\Delta} \in \mathbb{R}^{d \times d}}{\operatorname{argmin}} \ell(\boldsymbol{\Delta}) + \mathcal{G}_\lambda(\boldsymbol{\Delta}), \tag{A.1}$$

where $\ell(\boldsymbol{\Delta}) = 1/2 \operatorname{tr}(\boldsymbol{\Delta} \widehat{\boldsymbol{\Sigma}}_Y \boldsymbol{\Delta} \widehat{\boldsymbol{\Sigma}}_X) - \operatorname{tr}\big(\boldsymbol{\Delta}(\widehat{\boldsymbol{\Sigma}}_X - \widehat{\boldsymbol{\Sigma}}_Y)\big)$, $\mathcal{G}_\lambda(\boldsymbol{\Delta})$ is the nonconvex penalty defined in Section 3 and $\lambda$ is a non-negative regularization parameter. By the definition and decomposition of nonconvex penalty in Section 3, we can written the estimator as

$$\widehat{\boldsymbol{\Delta}} = \underset{\boldsymbol{\Delta} \in \mathbb{R}^{d \times d}}{\operatorname{argmin}} \widetilde{\ell}_\lambda(\boldsymbol{\Delta}) + \lambda \|\boldsymbol{\Delta}\|_{1,1}, \tag{A.2}$$

where $\widetilde{\ell}_\lambda(\boldsymbol{\Delta}) = \ell(\boldsymbol{\Delta}) + \mathcal{H}_\lambda(\boldsymbol{\Delta})$, and $\mathcal{H}_\lambda(\boldsymbol{\Delta}) = \sum_{j,k=1}^d h_\lambda(\Delta_{jk})$ is the concave part of $\mathcal{G}(\boldsymbol{\Delta})$.

To simplify the proof, we further make some transformations on the notations. By some linear algebra identities [13], we have $\operatorname{tr}(\mathbf{A}^\top \mathbf{B}) = \operatorname{vec}(\mathbf{A})^\top \operatorname{vec}(\mathbf{B})$ and $\operatorname{tr}(\mathbf{A}^\top \mathbf{B} \mathbf{C} \mathbf{D}^\top) = \operatorname{vec}(\mathbf{A})^\top (\mathbf{D} \otimes \mathbf{B}) \operatorname{vec}(\mathbf{C})$ for any matrices $\mathbf{A}, \mathbf{B}, \mathbf{C}$ and $\mathbf{D}$ with commensurate dimensions. Using these identities, we can rewrite the quasi log likelihood in (3.6) as

$$\mathcal{L}(\boldsymbol{\beta}) = \frac{1}{2}\boldsymbol{\beta}^\top \widehat{\mathbf{Q}} \boldsymbol{\beta} - \widehat{\mathbf{b}}^\top \boldsymbol{\beta}, \tag{A.3}$$

where $\boldsymbol{\beta} = \operatorname{vec}(\boldsymbol{\Delta}) \in \mathbb{R}^{d^2}$, $\widehat{\mathbf{Q}} = \widehat{\boldsymbol{\Sigma}}_X \otimes \widehat{\boldsymbol{\Sigma}}_Y \in \mathbb{R}^{d^2 \times d^2}$ and $\widehat{\mathbf{b}} = \operatorname{vec}(\widehat{\boldsymbol{\Sigma}}_X - \widehat{\boldsymbol{\Sigma}}_Y) \in \mathbb{R}^{d^2}$. Then the estimator in (A.1) can be rewritten as

$$\widehat{\boldsymbol{\beta}} = \underset{\boldsymbol{\beta} \in \mathbb{R}^{d^2}}{\operatorname{argmin}} \mathcal{L}(\boldsymbol{\beta}) + \mathcal{G}_\lambda(\boldsymbol{\beta}), \tag{A.4}$$

where $\mathcal{L}(\boldsymbol{\beta}) = 1/2 \boldsymbol{\beta}^\top \widehat{\mathbf{Q}} \boldsymbol{\beta} - \widehat{\mathbf{b}}^\top \boldsymbol{\beta}$ is the counterpart of loss function $\ell(\boldsymbol{\Delta}) = 1/2 \operatorname{tr}(\boldsymbol{\Delta} \widehat{\boldsymbol{\Sigma}}_Y \boldsymbol{\Delta} \widehat{\boldsymbol{\Sigma}}_X) - \operatorname{tr}\big(\boldsymbol{\Delta}(\widehat{\boldsymbol{\Sigma}}_X - \widehat{\boldsymbol{\Sigma}}_Y)\big)$, $\mathcal{G}_\lambda(\boldsymbol{\beta}) = \sum_{i=1}^{d^2} g_\lambda(\beta_i)$ is the nonconvex penalty defined in Section 3 and $\lambda$ is a non-negative regularization parameter. Therefore, the optimization problem in (A.2) turns to be

$$\widehat{\boldsymbol{\beta}} = \underset{\boldsymbol{\beta} \in \mathbb{R}^{d^2}}{\operatorname{argmin}} \widetilde{\mathcal{L}}_\lambda(\boldsymbol{\beta}) + \lambda \|\boldsymbol{\beta}\|_1, \tag{A.5}$$

where $\widetilde{\mathcal{L}}_\lambda(\boldsymbol{\beta}) = \mathcal{L}(\boldsymbol{\beta}) + \mathcal{H}_\lambda(\boldsymbol{\beta})$, and $\mathcal{H}_\lambda(\boldsymbol{\beta}) = \sum_{i=1}^{d^2} h_\lambda(\beta_i)$ is the concave part of $\mathcal{G}(\boldsymbol{\beta})$.

Denote $\operatorname{vec}(S) := \operatorname{supp}(\boldsymbol{\beta}^*)$, where $\boldsymbol{\beta}^* = \operatorname{vec}(\boldsymbol{\Delta}^*)$ and $S = \operatorname{supp}(\boldsymbol{\Delta}^*)$ is the support of the true differential graph. Finally, the vectorized oracle estimator of $\boldsymbol{\beta}^*$ in (4.1) turns to be

$$\widehat{\boldsymbol{\beta}}_O = \underset{\operatorname{supp}(\boldsymbol{\beta}) \subseteq \operatorname{vec}(S)}{\operatorname{argmin}} \mathcal{L}(\boldsymbol{\beta}), \tag{A.6}$$

where $\mathcal{L}(\boldsymbol{\beta}) = \frac{1}{2}\boldsymbol{\beta}^\top \widehat{\mathbf{Q}} \boldsymbol{\beta} - \widehat{\mathbf{b}}^\top \boldsymbol{\beta}$.

Now, we are ready to prove our main results. In order to make the proof concise, we first prove Theorem 4.6, followed which we prove Theorem 4.4. Note that the proof of Theorem 4.4 relies on the proof of Theorem 4.6.

*Proof of Theorem 4.6.* Suppose $\widehat{\mathbf{z}} \in \partial \|\widehat{\boldsymbol{\beta}}\|_1$. In particular, the estimator $\widehat{\boldsymbol{\beta}}$ in (A.5) satisfies optimality condition for unconstrained problem

$$\langle \widehat{\boldsymbol{\beta}} - \boldsymbol{\beta}', \nabla \widetilde{\mathcal{L}}_\lambda(\widehat{\boldsymbol{\beta}}) + \lambda \widehat{\mathbf{z}} \rangle \leq 0, \tag{A.7}$$

for any $\boldsymbol{\beta}'$.

First, we want to show that there exists some $\widehat{\mathbf{z}}_O \in \partial \|\widehat{\boldsymbol{\beta}}_O\|_1$, such that $\widehat{\mathbf{z}}_O$ satisfies the optimality condition as follows

$$\langle \widehat{\boldsymbol{\beta}}_O - \boldsymbol{\beta}', \nabla \widetilde{\mathcal{L}}_\lambda(\widehat{\boldsymbol{\beta}}_O) + \lambda \widehat{\mathbf{z}}_O \rangle \leq 0, \tag{A.8}$$

for any $\boldsymbol{\beta}'$. Since $\widetilde{\mathcal{L}}_\lambda(\boldsymbol{\beta}) = \mathcal{L}(\boldsymbol{\beta}) + \mathcal{H}_\lambda(\boldsymbol{\beta})$, we have

$$\langle \widehat{\boldsymbol{\beta}}_O - \boldsymbol{\beta}', \nabla\widetilde{\mathcal{L}}_\lambda(\widehat{\boldsymbol{\beta}}_O) + \lambda\widehat{\mathbf{z}}_O \rangle = \underbrace{\sum_{i \in \mathrm{vec}(S)} (\widehat{\boldsymbol{\beta}}_O - \boldsymbol{\beta}')_i \cdot \left(\nabla\widetilde{\mathcal{L}}_\lambda(\widehat{\boldsymbol{\beta}}_O) + \lambda\widehat{\mathbf{z}}_O\right)_i}_{(i)}$$

$$+ \underbrace{\sum_{i \in \mathrm{vec}(S)^c} (\widehat{\boldsymbol{\beta}}_O - \boldsymbol{\beta}')_i \cdot \left(\nabla\widetilde{\mathcal{L}}_\lambda(\widehat{\boldsymbol{\beta}}_O) + \lambda\widehat{\mathbf{z}}_O\right)_i}_{(ii)}. \qquad \text{(A.9)}$$

For term (i) in (A.9), by Lemma B.3, we have with probability at least $1 - 3/s$ that

$$\|\widehat{\boldsymbol{\beta}}_O - \boldsymbol{\beta}^*\|_\infty \leq C\theta_X^2\theta_Y^2\sigma_X\sigma_Y M\sqrt{\frac{\log s}{n}},$$

where $C$ is an absolute constant. Recall the assumption on entry magnitude of $\boldsymbol{\beta}^*$, *i.e.*, $\min_{i \in \mathrm{vec}(S)} |\beta_i^*| \geq \nu + C\theta_X^2\theta_Y^2\sigma_X\sigma_Y M\sqrt{\log s/n}$, we have with probability at least $1 - 3/s$ that

$$\min_{i \in \mathrm{vec}(S)} \left|(\widehat{\boldsymbol{\beta}}_O)_i\right| = \min_{i \in \mathrm{vec}(S)} \left|(\widehat{\boldsymbol{\beta}}_O - \boldsymbol{\beta}^* + \boldsymbol{\beta}^*)_i\right| \geq \min_{i \in \mathrm{vec}(S)} \left\{\left|(\boldsymbol{\beta}^*)_i\right| - \left|(\widehat{\boldsymbol{\beta}}_O - \boldsymbol{\beta}^*)_i\right|\right\}$$

$$\geq - \max_{i \in \mathrm{vec}(S)} \left|(\widehat{\boldsymbol{\beta}}_O - \boldsymbol{\beta}^*)_i\right| + \min_{i \in \mathrm{vec}(S)} \left|(\boldsymbol{\beta}^*)_i\right|. \qquad \text{(A.10)}$$

The right hand side of (A.10) can be further lower bounded by

$$\min_{i \in S} \left|(\widehat{\boldsymbol{\beta}}_O)_i\right| \geq -C\theta_X^2\theta_Y^2\sigma_X\sigma_Y M\sqrt{\frac{\log s}{n}} + \nu + C\theta_X^2\theta_Y^2\sigma_X\sigma_Y M\sqrt{\frac{\log s}{n}}.$$

Following condition (a) in Assumption 4.3 for $\mathcal{G}(\boldsymbol{\beta})$, we have

$$(\nabla\mathcal{H}_\lambda(\widehat{\boldsymbol{\beta}}_O) + \lambda\widehat{\mathbf{z}}_O)_i = (\nabla\mathcal{G}(\widehat{\boldsymbol{\beta}}_O))_i = g_\lambda'((\widehat{\boldsymbol{\beta}}_O)_i) = 0,$$

for $i \in \mathrm{vec}(S)$. Hence we have

$$\sum_{i \in \mathrm{vec}(S)} (\widehat{\boldsymbol{\beta}}_O - \boldsymbol{\beta}')_i (\nabla\widetilde{\mathcal{L}}_\lambda(\widehat{\boldsymbol{\beta}}_O) + \lambda\widehat{\mathbf{z}}_O)_i = \sum_{i \in \mathrm{vec}(S)} (\widehat{\boldsymbol{\beta}}_O - \boldsymbol{\beta}')_i \cdot \left(\nabla\mathcal{L}(\widehat{\boldsymbol{\beta}}_O) + \nabla\mathcal{H}_\lambda(\widehat{\boldsymbol{\beta}}_O) + \lambda\widehat{\mathbf{z}}_O\right)_i,$$

$$= \sum_{i \in \mathrm{vec}(S)} (\widehat{\boldsymbol{\beta}}_O - \boldsymbol{\beta}')_i \cdot \left(\nabla\mathcal{L}(\widehat{\boldsymbol{\beta}}_O)\right)_i.$$

Recall that $\widehat{\boldsymbol{\beta}}_O$ is the global solution to the problem in (A.6). Hence we have $\widehat{\boldsymbol{\beta}}_O$ satisfies the optimality condition as follows

$$\sum_{i \in \mathrm{vec}(S)} (\widehat{\boldsymbol{\beta}}_O - \boldsymbol{\beta}')_i \left(\nabla\mathcal{L}(\widehat{\boldsymbol{\beta}}_O)\right)_i \leq 0,$$

which leads to

$$\sum_{i \in \mathrm{vec}(S)} (\widehat{\boldsymbol{\beta}}_O - \boldsymbol{\beta}')_i \left(\nabla\widetilde{\mathcal{L}}_\lambda(\widehat{\boldsymbol{\beta}}_O) + \lambda\widehat{\mathbf{z}}_O\right)_i \leq 0. \qquad \text{(A.11)}$$

For term (ii) in (A.9), notice that $(\widehat{\boldsymbol{\beta}}_O)_i = 0$ for $i \in \mathrm{vec}(S)^c$. By the regularity condition (c), we have

$$\left(\nabla\mathcal{H}_\lambda(\widehat{\boldsymbol{\beta}}_O)\right)_i = h_\lambda'((\widehat{\boldsymbol{\beta}}_O)_i) = 0,$$

for $i \in \mathrm{vec}(S)^c$. This leads to

$$\sum_{i \in \mathrm{vec}(S)^c} (\widehat{\boldsymbol{\beta}}_O - \boldsymbol{\beta}')_i \cdot \left(\nabla\widetilde{\mathcal{L}}_\lambda(\widehat{\boldsymbol{\beta}}_O) + \lambda\widehat{\mathbf{z}}_O\right)_i = \sum_{i \in \mathrm{vec}(S)^c} (\widehat{\boldsymbol{\beta}}_O - \boldsymbol{\beta}')_i \cdot (\nabla\mathcal{L}(\widehat{\boldsymbol{\beta}}_O) + \nabla\mathcal{H}_\lambda(\widehat{\boldsymbol{\beta}}_O) + \lambda\widehat{\mathbf{z}}_O)_i,$$

$$= \sum_{i \in \mathrm{vec}(S)^c} (\widehat{\boldsymbol{\beta}}_O - \boldsymbol{\beta}')_i \cdot (\nabla\mathcal{L}(\widehat{\boldsymbol{\beta}}_O) + \lambda\widehat{\mathbf{z}}_O)_i.$$

Since $\nabla\mathcal{L}(\boldsymbol{\beta}) = \widehat{\mathbf{Q}}\boldsymbol{\beta} - \widehat{\mathbf{b}}$ and note that $\mathbf{Q}^*\boldsymbol{\beta}^* = \mathbf{b}^*$, we have

$$
\begin{aligned}
\left\|\nabla\mathcal{L}(\widehat{\boldsymbol{\beta}}_O)\right\|_\infty &= \left\|\widehat{\mathbf{Q}}\widehat{\boldsymbol{\beta}}_O - \widehat{\mathbf{Q}}\boldsymbol{\beta}^* + \widehat{\mathbf{Q}}\boldsymbol{\beta}^* - \mathbf{Q}^*\boldsymbol{\beta}^* + \mathbf{b}^* - \widehat{\mathbf{b}}\right\|_\infty \\
&\leq \left\|\widehat{\mathbf{Q}}\right\|_1 \cdot \left\|\widehat{\boldsymbol{\beta}}_O - \boldsymbol{\beta}^*\right\|_\infty + \left\|\widehat{\mathbf{Q}} - \mathbf{Q}^*\right\|_{\infty,\infty} \cdot \left\|\boldsymbol{\beta}^*\right\|_1 + \left\|\mathbf{b}^* - \widehat{\mathbf{b}}\right\|_\infty \\
&\leq \left\|\widehat{\mathbf{Q}}\right\|_1 \cdot C\theta_X^2\theta_Y^2\sigma_X\sigma_Y M\sqrt{\frac{\log s}{n}} + \left\|\boldsymbol{\beta}^*\right\|_1 \cdot \sqrt{5}\pi\sqrt{\frac{\log d}{n}} + 6\pi\sqrt{\frac{\log d}{n}},
\end{aligned}
$$

where in the last inequality the first term is due to Lemma B.3, the second term is from (B.3), and the last term is due to Lemma C.1 and (B.6). In addition, we have $\|\widehat{\mathbf{Q}}\|_1 \leq \|\widehat{\boldsymbol{\Sigma}}_X\|_1 \cdot \|\widehat{\boldsymbol{\Sigma}}_Y\|_1 \leq 4\|\boldsymbol{\Sigma}_X^*\|_1 \cdot \|\boldsymbol{\Sigma}_Y^*\|_1$ when $n$ is sufficient large, and thus $\|\widehat{\mathbf{Q}}\|_1 \leq 4\sigma_X\sigma_Y$ by Assumption 4.1. By Assumption 4.2, we have $\|\boldsymbol{\beta}^*\|_1 \leq M$. Therefore, for any $i \in \mathrm{vec}(S)^c$, we obtain

$$
\left|\left(\nabla\mathcal{L}(\widehat{\boldsymbol{\beta}}_O)\right)_i\right| \leq \left\|\nabla\mathcal{L}(\widehat{\boldsymbol{\beta}}_O)\right\|_\infty \leq C_0\theta_X^2\theta_Y^2\sigma_X\sigma_Y M\sqrt{\frac{\log d}{n}},
$$

where $C_0$ is an absolute constant. By Assumption , it follows that $\left|\left(\nabla\mathcal{L}(\widehat{\boldsymbol{\beta}}_O)\right)_i\right| \leq \lambda/2$ for any $i \in \mathrm{vec}(S)^c$. Since we have $\widehat{\mathbf{z}}_O \in \partial\|\widehat{\boldsymbol{\beta}}_O\|_1$, hence $|(\widehat{\mathbf{z}}_O)_i| \leq 1$ for $i \in \mathrm{vec}(S)^c$. By setting $(\widehat{\mathbf{z}}_O)_i = -(\nabla\mathcal{L}(\widehat{\boldsymbol{\beta}}_O))_i/\lambda$ for $i \in \mathrm{vec}(S)^c$, we can enforce the following equality to hold

$$
\left(\nabla\mathcal{L}(\widehat{\boldsymbol{\beta}}_O) + \lambda\widehat{\mathbf{z}}_O\right)_i = 0,
$$

for $i \in \mathrm{vec}(S)^c$. Hence, we have

$$
\sum_{i\in\mathrm{vec}(S)^c} (\widehat{\boldsymbol{\beta}}_O - \boldsymbol{\beta}')_i \cdot \left(\nabla\widetilde{\mathcal{L}}_\lambda(\widehat{\boldsymbol{\beta}}_O) + \lambda\widehat{\mathbf{z}}_O\right)_i = \sum_{i\in\mathrm{vec}(S)^c} (\widehat{\boldsymbol{\beta}}_O - \boldsymbol{\beta}')_i \cdot \left(\nabla\mathcal{L}(\widehat{\boldsymbol{\beta}}_O) + \lambda\widehat{\mathbf{z}}_O\right)_i = 0.
$$

(A.12)

By using (A.11) and (A.12), we obtain (A.8).

Now we are ready to provide proof on $\widehat{\boldsymbol{\beta}} = \widehat{\boldsymbol{\beta}}_O$. Recall that $\mathrm{supp}(\widehat{\boldsymbol{\beta}}_O) = \mathrm{vec}(S)$, and Lemma B.2 shows that under suitable condition, we have

$$
\widetilde{\mathcal{L}}_\lambda(\widehat{\boldsymbol{\beta}}) \geq \widetilde{\mathcal{L}}_\lambda(\widehat{\boldsymbol{\beta}}_O) + \langle\nabla\widetilde{\mathcal{L}}_\lambda(\widehat{\boldsymbol{\beta}}_O), \widehat{\boldsymbol{\beta}} - \widehat{\boldsymbol{\beta}}_O\rangle + \frac{\rho - \zeta_-}{2}\|\widehat{\boldsymbol{\beta}} - \widehat{\boldsymbol{\beta}}_O\|_2^2, \tag{A.13}
$$

$$
\widetilde{\mathcal{L}}_\lambda(\widehat{\boldsymbol{\beta}}_O) \geq \widetilde{\mathcal{L}}_\lambda(\widehat{\boldsymbol{\beta}}) + \langle\nabla\widetilde{\mathcal{L}}_\lambda(\widehat{\boldsymbol{\beta}}), \widehat{\boldsymbol{\beta}}_O - \widehat{\boldsymbol{\beta}}\rangle + \frac{\rho - \zeta_-}{2}\|\widehat{\boldsymbol{\beta}}_O - \widehat{\boldsymbol{\beta}}\|_2^2, \tag{A.14}
$$

hold with high probability.

By convexity of $\ell_1$ norm $\|\cdot\|_1$, we have following two inequality hold

$$
\lambda\|\widehat{\boldsymbol{\beta}}\|_1 \geq \lambda\|\widehat{\boldsymbol{\beta}}_O\|_1 + \lambda\langle\widehat{\boldsymbol{\beta}} - \widehat{\boldsymbol{\beta}}_O, \widehat{\mathbf{z}}_O\rangle, \tag{A.15}
$$

$$
\lambda\|\widehat{\boldsymbol{\beta}}_O\|_1 \geq \lambda\|\widehat{\boldsymbol{\beta}}\|_1 + \lambda\langle\widehat{\boldsymbol{\beta}}_O - \widehat{\boldsymbol{\beta}}, \widehat{\mathbf{z}}\rangle. \tag{A.16}
$$

By adding Equations (A.13)-(A.16), we have

$$
0 \geq \underbrace{\langle\widehat{\boldsymbol{\beta}}_O - \widehat{\boldsymbol{\beta}}, \nabla\widetilde{\mathcal{L}}_\lambda(\widehat{\boldsymbol{\beta}}) + \lambda\widehat{\mathbf{z}}\rangle}_{(a)} + \underbrace{\langle\widehat{\boldsymbol{\beta}} - \widehat{\boldsymbol{\beta}}_O, \nabla\widetilde{\mathcal{L}}_\lambda(\widehat{\boldsymbol{\beta}}_O) + \lambda\widehat{\mathbf{z}}_O\rangle}_{(b)} + (\rho - \zeta_-)\|\widehat{\boldsymbol{\beta}} - \widehat{\boldsymbol{\beta}}_O\|_2^2.
$$

Recall that $\widehat{\boldsymbol{\beta}}$ satisfies the optimality condition

$$
\langle\widehat{\boldsymbol{\beta}} - \widehat{\boldsymbol{\beta}}_O, \nabla\widetilde{\mathcal{L}}_\lambda(\widehat{\boldsymbol{\beta}}) + \lambda\widehat{\mathbf{z}}\rangle \leq 0,
$$

hence we have term (a) $\geq 0$.

Similarly, by (A.8), we have

$$
\langle\widehat{\boldsymbol{\beta}}_O - \widehat{\boldsymbol{\beta}}, \nabla\widetilde{\mathcal{L}}_\lambda(\widehat{\boldsymbol{\beta}}_O) + \lambda\widehat{\mathbf{z}}_O\rangle \leq 0,
$$

which leads to term (b) $\geq 0$. Therefore, we have $(\rho - \zeta_-)\|\widehat{\boldsymbol{\beta}} - \widehat{\boldsymbol{\beta}}_O\|_2^2 \leq 0$, which implies $\widehat{\boldsymbol{\beta}} = \widehat{\boldsymbol{\beta}}_O$. Thus we can conclude that, under suitable condition, the proposed estimator $\widehat{\boldsymbol{\beta}}$ is the oracle estimator $\widehat{\boldsymbol{\beta}}_O$, which exactly recover the true support of $\boldsymbol{\beta}^*$ with probability at least $1 - 3/s$. $\qquad\square$

Next, we are able to prove Theorem 4.4.

*Proof of Theorem 4.4.* Recall that in Theorem 4.6, we have proved that under certain conditions $\widehat{\boldsymbol{\beta}} = \widehat{\boldsymbol{\beta}}_O$ holds. Then by Lemma B.3 we have

$$\left\|\widehat{\boldsymbol{\beta}} - \boldsymbol{\beta}^*\right\|_\infty = \left\|\widehat{\boldsymbol{\beta}}_O - \boldsymbol{\beta}^*\right\|_\infty \leq 6\pi\theta_X\theta_Y\sqrt{\frac{\log s}{n}} + 2\sqrt{10}\pi\theta_X^2\theta_Y^2\sigma_X\sigma_Y M\sqrt{\frac{\log s}{n}}$$

holds with probability at least $1 - 3/s$, where the second term dominates the first one.

Next, we just need to bound $\left\|\widehat{\boldsymbol{\beta}}_O - \boldsymbol{\beta}^*\right\|_2$. By definition in (A.6),we have

$$\widehat{\boldsymbol{\beta}}_O = \underset{\mathrm{supp}(\boldsymbol{\beta})\subseteq\mathrm{vec}(S)}{\mathrm{argmin}} \frac{1}{2}\boldsymbol{\beta}^\top\widehat{\mathbf{Q}}\boldsymbol{\beta} - \widehat{\mathbf{b}}^\top\boldsymbol{\beta}.$$

By definition we have $\widehat{\mathbf{Q}} = \widehat{\boldsymbol{\Sigma}}_X \otimes \widehat{\boldsymbol{\Sigma}}_Y \in \mathbb{R}^{d^2 \times d^2}$. For any $j, k, p, q = 1, \ldots, d$, we use $\widehat{\mathbf{Q}}_{(j,k,p,q)}$ to denote the entry in $\widehat{\mathbf{Q}}$ that is obtained from the product of the $(j, k)$-th entry in $\widehat{\boldsymbol{\Sigma}}_X$ and the $(p, q)$-th entry in $\widehat{\boldsymbol{\Sigma}}_Y$. Specifically, we have

$$\widehat{\mathbf{Q}}_{(j,k,p,q)} = \widehat{\Sigma}_X^{jk}\widehat{\Sigma}_Y^{pq} = \sin\left(\frac{\pi}{2}\widehat{\tau}_{jk}^X\right)\sin\left(\frac{\pi}{2}\widehat{\tau}_{pq}^Y\right)$$

$$= \frac{1}{2}\cos\left(\frac{\pi}{2}\left(\widehat{\tau}_{jk}^X - \widehat{\tau}_{pq}^Y\right)\right) - \frac{1}{2}\cos\left(\frac{\pi}{2}\left(\widehat{\tau}_{jk}^X + \widehat{\tau}_{pq}^Y\right)\right).$$

Furthermore, we define $\widehat{\mu}_{jk;pq} = \widehat{\tau}_{jk}^X - \widehat{\tau}_{pq}^Y$ and $\widehat{\mu}'_{jk;pq} = \widehat{\tau}_{jk}^X + \widehat{\tau}_{pq}^Y$. All the notations above can be easily extended to $\mathbf{Q}^*$. Then we have

$$\widehat{\mathbf{Q}}_{(j,k,p,q)} - \mathbf{Q}^*_{(j,k,p,q)} = \frac{1}{2}\left(\cos\left(\frac{\pi}{2}\widehat{\mu}_{jk;pq}\right) - \cos\left(\frac{\pi}{2}\mu^*_{jk;pq}\right)\right)$$

$$+ \frac{1}{2}\left(\cos\left(\frac{\pi}{2}\widehat{\mu}'_{jk;pq}\right) - \cos\left(\frac{\pi}{2}\mu'^*_{jk;pq}\right)\right).$$

We only need to bound the first term, and the second term is very similar and the bound should be exactly the same. Note that

$$\cos\left(\frac{\pi}{2}\widehat{\mu}_{jk;pq}\right) - \cos\left(\frac{\pi}{2}\mu^*_{jk;pq}\right) = -\frac{\pi}{2}\sin\left(\frac{\pi}{2}\mu^*_{jk;pq}\right)\left(\widehat{\mu}_{jk;pq} - \mu^*_{jk;pq}\right)$$

$$- \frac{\pi^2}{8}\cos\left(\frac{\pi}{2}\widetilde{\mu}_{jk;pq}\right)\left(\widehat{\mu}_{jk;pq} - \mu^*_{jk;pq}\right)^2,$$

where $\widetilde{\mu}_{jk;pq}$ lies between $\widehat{\mu}_{jk;pq}$ and $\mu^*_{jk;pq}$. Let $\widehat{\mathbf{L}} \in \mathbb{R}^{d^2 \times d^2}$ be the matrix with the same structure as $\widehat{\mathbf{Q}}$ whose $(j, k, p, q)$-th entry is $\cos(\pi/2\widehat{\mu}_{jk;pq})$. Similar notations are defined for $\mathbf{L}^*$ and $\widetilde{\mathbf{L}}$. Then for any $\mathbf{x} \in \mathbb{R}^{d^2}$ we have

$$\left|\mathbf{x}^\top\left(\widehat{\mathbf{Q}} - \mathbf{Q}^*\right)\mathbf{x}\right| \leq \frac{\pi}{2}\left|\mathbf{x}^\top\left[\sin\left(\frac{\pi}{2}\widehat{\mathbf{L}}\right)\circ\left(\widehat{\mathbf{L}} - \mathbf{L}^*\right)\right]\mathbf{x}\right| + \frac{\pi^2}{8}\left|\mathbf{x}^\top\left[\cos\left(\frac{\pi}{2}\widetilde{\mathbf{L}}\right)\circ\left(\widehat{\mathbf{L}} - \mathbf{L}^*\right)\circ\left(\widehat{\mathbf{L}} - \mathbf{L}^*\right)\right]\mathbf{x}\right|.$$

Recall the results in Lemma C.2 and Lemma C.3, following a similar proof we can show that with probability at least $1 - 1/s$

$$\sup_{\|\mathbf{x}\|\leq 1}\left|\mathbf{x}^\top\left(\widehat{\mathbf{Q}}_{SS} - \mathbf{Q}^*_{SS}\right)\mathbf{x}\right| \leq 2\pi^2\frac{s\log s}{n} + 8\pi\sqrt{C}\sqrt{\frac{\log s + s\log 9}{n}}, \quad (A.17)$$

where $C$ is an absolute constant. We have the closed form solution $\boldsymbol{\beta}^* = \mathbf{Q}^{*-1}\mathbf{b}^*$. It follows that

$$\left\|\widehat{\boldsymbol{\beta}}_O - \boldsymbol{\beta}^*\right\|_2 = \left\|\widehat{\mathbf{Q}}_{SS}^{-1}\widehat{\mathbf{b}} - \mathbf{Q}^{*-1}\mathbf{b}^*\right\|_2 = \left\|\widehat{\mathbf{Q}}_{SS}^{-1}\widehat{\mathbf{b}} - \widehat{\mathbf{Q}}_{SS}^{-1}\mathbf{b}^* + \widehat{\mathbf{Q}}_{SS}^{-1}\mathbf{b}^* - \mathbf{Q}^{*-1}\mathbf{b}^*\right\|_2$$

$$\leq \underbrace{\left\|\widehat{\mathbf{Q}}_{SS}^{-1}\right\|_2 \cdot \left\|\left[\widehat{\mathbf{b}} - \mathbf{b}^*\right]_S\right\|_2}_{(i)} + \underbrace{\left\|\widehat{\mathbf{Q}}_{SS}^{-1} - \mathbf{Q}^{*-1}_{SS}\right\|_2 \cdot \left\|\mathbf{b}^*_S\right\|_2}_{(ii)},$$

$$(A.18)$$

where we use the fact that $\mathrm{vec}(S) = \mathrm{supp}(\boldsymbol{\beta}^*) = \mathrm{supp}(\mathbf{Q}^{*-1}\mathbf{b}^*)$. Note that $\|\mathbf{b}^*\|_2 = \|\boldsymbol{\Sigma}_X^* - \boldsymbol{\Sigma}_Y^*\|_F = \|\boldsymbol{\Sigma}_X^*\boldsymbol{\Delta}^*\boldsymbol{\Sigma}_Y^*\|_F \leq \|\boldsymbol{\Sigma}_X^*\|_2 \cdot \|\boldsymbol{\Sigma}_X^*\|_2 \cdot \|\boldsymbol{\Delta}^*\|_F \leq M/(\kappa_1\kappa_2)$, here we used the fact that

$\|\mathbf{\Delta}^*\|_F \leq \|\mathbf{\Delta}^*\|_{1,1} \leq M$ and $\lambda_{\max}(\mathbf{\Sigma}_X^*) \leq 1/\kappa_1$ by Assumption 4.1 and 4.2. Then term (ii) in (A.18) can be bounded as

$$\left\|\widehat{\mathbf{Q}}_{SS}^{-1} - \mathbf{Q}_{SS}^{*-1}\right\|_2 \cdot \|\mathbf{b}_S^*\|_2 \leq \frac{M}{\kappa_1 \kappa_2} \cdot \left\|\widehat{\mathbf{Q}}_{SS}^{-1}\right\|_2 \cdot \left\|\widehat{\mathbf{Q}}_{SS} - \mathbf{Q}_{SS}^*\right\|_2 \cdot \left\|\mathbf{Q}_{SS}^{*-1}\right\|_2$$

$$\leq \frac{4\pi^2 M}{\kappa_1^3 \kappa_2^3} \frac{s \log s}{n} + \frac{16\pi MC}{\kappa_1^3 \kappa_2^3} \sqrt{\frac{\log s + s \log 9}{n}}, \qquad (A.19)$$

where the second inequality uses the bound in (A.17) and the fact that $\left\|\mathbf{Q}_{SS}^{*-1}\right\|_2 \leq 1/(\kappa_1 \kappa_2)$ by Assumption 4.1 and $\left\|\widehat{\mathbf{Q}}_{SS}^{-1}\right\|_2 \leq 2\left\|\mathbf{Q}_{SS}^{*-1}\right\|_2$ when $n$ is sufficient large.

For term (i) in (A.18), with probability at least $1 - 1/d - 2/d^2$ we have $\|\widehat{\mathbf{\Sigma}}_X - \mathbf{\Sigma}_X^*\|_2 \leq 2\pi^2 d \log d/n + 8\pi\sqrt{C}\sqrt{(\log d + d \log 9)/n}$ by Lemma C.3. The dominating term is $\sqrt{d/n}$. Similar bound for $\widehat{\mathbf{\Sigma}}_Y$ holds. It immediately implies

$$\left\|\widehat{\mathbf{Q}}_{SS}^{-1}\right\|_2 \cdot \left\|\left[\widehat{\mathbf{b}} - \mathbf{b}^*\right]_S\right\|_2 \leq \frac{2}{\kappa_1 \kappa_2}\left(\left\|\left[\widehat{\mathbf{\Sigma}}_X - \mathbf{\Sigma}_X^*\right]_S\right\|_2 + \left\|\left[\widehat{\mathbf{\Sigma}}_Y - \mathbf{\Sigma}_Y^*\right]_S\right\|_2\right) \leq \frac{C_1}{\kappa_1 \kappa_2}\sqrt{\frac{s}{n}}, \tag{A.20}$$

where $C_1$ is an absolute constant. Submitting (A.19) and (A.20) into (A.18), we obtain

$$\left\|\widehat{\boldsymbol{\beta}}_O - \boldsymbol{\beta}^*\right\|_2 \leq \frac{C_1}{\kappa_1 \kappa_2}\sqrt{\frac{s}{n}} + \frac{4\pi^2 M}{\kappa_1^3 \kappa_2^3}\frac{s \log s}{n} + \frac{64(1 + \sqrt{5})\pi M}{\kappa_1^3 \kappa_2^3}\sqrt{\frac{\log s + s \log 9}{n}},$$

which holds with probability at least $1 - 1/d - 1/d^{2.5}$. In Theorem 4.6, we have proved that under certain conditions $\widehat{\boldsymbol{\beta}} = \widehat{\boldsymbol{\beta}}_O$ holds, which further implies that

$$\left\|\widehat{\boldsymbol{\beta}} - \boldsymbol{\beta}^*\right\|_2 \leq \frac{C_1(\kappa_1^2 + \kappa_2^2)}{\kappa_1^3 \kappa_2^3}\sqrt{\frac{s}{n}} + \frac{4\pi^2 \sigma_X \sigma_Y M}{\kappa_1^3 \kappa_2^3}\frac{s \log s}{n} + \frac{64(1 + \sqrt{5})\pi M}{\kappa_1^3 \kappa_2^3}\sqrt{\frac{\log s + s \log 9}{n}}$$

$$\leq \frac{C_2 M}{\kappa_1 \kappa_2}\sqrt{\frac{s}{n}}$$

holds with probability at least $1 - 1/s - 1/s^{2.5} \geq 1 - 2/s$, where $C_2$ is an absolute constant. $\qquad\square$

Finally, we are ready to prove Theorem 4.7.

*Proof of Theorem 4.7.* Let $\mathbf{z} \in \partial\|\boldsymbol{\beta}\|_1$ and $\widehat{\mathbf{z}} \in \partial\|\widehat{\boldsymbol{\beta}}\|_1$ denote the subgradient. Recall that, $\widehat{\boldsymbol{\beta}}$ is the global solution to (A.5). Hence we have

$$\langle\widehat{\boldsymbol{\beta}} - \boldsymbol{\beta}', \nabla\widetilde{\mathcal{L}}_\lambda(\widehat{\boldsymbol{\beta}}) + \lambda\widehat{\mathbf{z}}\rangle \leq 0, \tag{A.21}$$

for any $\boldsymbol{\beta}'$. By Lemma B.2, under suitable condition, with high probability, we have

$$\widetilde{\mathcal{L}}_\lambda(\widehat{\boldsymbol{\beta}}) \geq \widetilde{\mathcal{L}}_\lambda(\boldsymbol{\beta}^*) + \langle\nabla\widetilde{\mathcal{L}}_\lambda(\boldsymbol{\beta}^*), \widehat{\boldsymbol{\beta}} - \boldsymbol{\beta}^*\rangle + \frac{\rho - \zeta_-}{2}\left\|\widehat{\boldsymbol{\beta}} - \boldsymbol{\beta}^*\right\|_2^2, \tag{A.22}$$

$$\widetilde{\mathcal{L}}_\lambda(\boldsymbol{\beta}^*) \geq \widetilde{\mathcal{L}}_\lambda(\widehat{\boldsymbol{\beta}}) + \langle\nabla\widetilde{\mathcal{L}}_\lambda(\widehat{\boldsymbol{\beta}}), \boldsymbol{\beta}^* - \widehat{\boldsymbol{\beta}}\rangle + \frac{\rho - \zeta_-}{2}\left\|\boldsymbol{\beta}^* - \widehat{\boldsymbol{\beta}}\right\|_2^2. \tag{A.23}$$

By convexity of $\ell_1$ norm $\|\cdot\|_1$, we have

$$\lambda\|\widehat{\boldsymbol{\beta}}\|_1 \geq \lambda\|\boldsymbol{\beta}^*\|_1 + \lambda\langle\widehat{\boldsymbol{\beta}} - \boldsymbol{\beta}^*, \mathbf{z}^*\rangle, \tag{A.24}$$

$$\lambda\|\boldsymbol{\beta}^*\|_1 \geq \lambda\|\widehat{\boldsymbol{\beta}}\|_1 + \lambda\langle\boldsymbol{\beta}^* - \widehat{\boldsymbol{\beta}}, \widehat{\mathbf{z}}\rangle. \tag{A.25}$$

Adding up (A.22) to (A.25), we have

$$0 \geq \langle\boldsymbol{\beta}^* - \widehat{\boldsymbol{\beta}}, \nabla\widetilde{\mathcal{L}}_\lambda(\widehat{\boldsymbol{\beta}}) + \lambda\widehat{\mathbf{z}}\rangle + \langle\widehat{\boldsymbol{\beta}} - \boldsymbol{\beta}^*, \nabla\widetilde{\mathcal{L}}_\lambda(\boldsymbol{\beta}^*) + \lambda\mathbf{z}^*\rangle + (\rho - \zeta_-)\|\widehat{\boldsymbol{\beta}} - \boldsymbol{\beta}^*\|_2^2.$$

Meanwhile, (A.21) leads to

$$\langle\nabla\widetilde{\mathcal{L}}_\lambda(\widehat{\boldsymbol{\beta}}) + \lambda\widehat{\mathbf{z}}, \boldsymbol{\beta}^* - \widehat{\boldsymbol{\beta}}\rangle \geq 0.$$

Hence, we have

$$(\rho - \zeta_-)\|\widehat{\boldsymbol{\beta}} - \boldsymbol{\beta}^*\|_2^2 \leq \langle \boldsymbol{\beta}^* - \widehat{\boldsymbol{\beta}}, \nabla\widetilde{\mathcal{L}}_\lambda(\boldsymbol{\beta}^*) + \lambda\mathbf{z}^* \rangle. \tag{A.26}$$

Recall that $\widetilde{\mathcal{L}}_\lambda(\boldsymbol{\beta})$ is restricted strongly convex provided $\rho = \kappa_1\kappa_2/2$. With $\zeta_- \leq \kappa_1\kappa_2/4$ and (A.26), we have

$$\begin{aligned}
\frac{\kappa_1\kappa_2}{4}\|\widehat{\boldsymbol{\beta}} - \boldsymbol{\beta}^*\|_2^2 &\leq (\rho - \zeta_-)\|\widehat{\boldsymbol{\beta}} - \boldsymbol{\beta}^*\|_2^2 \\
&\leq \langle \nabla\mathcal{L}(\boldsymbol{\beta}^*) + \nabla\mathcal{H}_\lambda(\boldsymbol{\beta}^*) + \lambda\mathbf{z}^*, \boldsymbol{\beta}^* - \widehat{\boldsymbol{\beta}} \rangle \\
&\leq \sum_{i=1}^{d^2} \left| \left( \nabla\mathcal{L}(\boldsymbol{\beta}^*) + \nabla\mathcal{H}_\lambda(\boldsymbol{\beta}^*) + \lambda\mathbf{z}^* \right)_i \right| \cdot \left| (\boldsymbol{\beta}^* - \widehat{\boldsymbol{\beta}})_i \right|. \tag{A.27}
\end{aligned}$$

Now, we decompose (A.27) into three parts: $i \in \mathrm{vec}(S)^c$, $i \in S_1$ and $i \in S_2$, where we define $S_1 = \{i \mid |(\boldsymbol{\beta}^*)_i| \geq \nu\}$ and $S_2 = \{i \mid |(\boldsymbol{\beta}^*)_i| < \nu\}$.

**Case 1:** For $i \in \mathrm{vec}(S)^c$, based on regularity condition (c) in Assumption 4.3, we have

$$\left( \nabla\mathcal{H}_\lambda(\boldsymbol{\beta}^*) \right)_i = h'_\lambda(\beta_i^*) = h'_\lambda(0) = 0.$$

Recall that we have $|\left( \nabla\mathcal{L}(\boldsymbol{\beta}^*) \right)_i| \leq CM\sqrt{\log d/n} = \lambda/2$ according to Lemma B.4. Hence,

$$\left| (\nabla\mathcal{L}(\boldsymbol{\beta}^*) + \nabla\mathcal{H}_\lambda(\boldsymbol{\beta}^*))_i \right| \leq \frac{\lambda}{2}.$$

Since $\mathbf{z}^* \in \partial\|\boldsymbol{\beta}^*\|_1$, we have $|z_i^*| \leq 1$ and thus $\lambda z_i^* \in [-\lambda, \lambda]$. Therefore, for any $i \in \mathrm{vec}(S)^c$, by definition of subgradient of $\mathbf{z}^*$ we can always find a $z_i^*$ such that

$$\left| (\nabla\mathcal{L}(\boldsymbol{\beta}^*) + \nabla\mathcal{H}_\lambda(\boldsymbol{\beta}^*) + \lambda\mathbf{z}^*)_i \right| = 0.$$

This leads to

$$\sum_{i \in \mathrm{vec}(S)^c} |\left( \nabla\mathcal{L}(\boldsymbol{\beta}^*) + \nabla\mathcal{H}_\lambda(\boldsymbol{\beta}^*) + \lambda\mathbf{z}^* \right)_i| \cdot |(\boldsymbol{\beta}^* - \widehat{\boldsymbol{\beta}})_i| = 0. \tag{A.28}$$

**Case 2:** For $i \in S_1$, we have $|\beta_i^*| \geq \nu$. By condition (a) in Assumption 4.3 on $\mathcal{G}(\boldsymbol{\beta}) = \mathcal{H}_\lambda(\boldsymbol{\beta}) + \lambda\|\boldsymbol{\beta}\|_1$, we have

$$\left( \nabla\mathcal{H}_\lambda(\boldsymbol{\beta}^*) + \lambda\mathbf{z}^* \right)_i = g'_\lambda(\beta_i^*) = 0,$$

which implies

$$\sum_{i \in S_1} |\left( \nabla\mathcal{L}(\boldsymbol{\beta}^*) + \nabla\mathcal{H}_\lambda(\boldsymbol{\beta}^*) + \lambda\mathbf{z}^* \right)_i| \cdot |(\boldsymbol{\beta}^* - \widehat{\boldsymbol{\beta}})_i| = \sum_{i \in S_1} |[\nabla\mathcal{L}(\boldsymbol{\beta}^*)]_i| \cdot |(\boldsymbol{\beta}^* - \widehat{\boldsymbol{\beta}})_i|.$$

Hence by Cauchy's inequality we have

$$\sum_{i \in S_1} |\left( \nabla\mathcal{L}(\boldsymbol{\beta}^*) + \nabla\mathcal{H}_\lambda(\boldsymbol{\beta}^*) + \lambda\mathbf{z}^* \right)_i| \cdot |(\boldsymbol{\beta}^* - \widehat{\boldsymbol{\beta}})_i| \leq \left\| \left( \nabla\mathcal{L}(\boldsymbol{\beta}^*) \right)_{S_1} \right\|_2 \cdot \|(\boldsymbol{\beta}^* - \widehat{\boldsymbol{\beta}})_{S_1}\|_2.$$

Since $\nabla\mathcal{L}(\boldsymbol{\beta}) = \widehat{\mathbf{Q}}\boldsymbol{\beta} - \widehat{\mathbf{b}}$ and note that $\mathbf{Q}^*\boldsymbol{\beta}^* = \mathbf{b}^*$, we have

$$\left\| [\nabla\mathcal{L}(\boldsymbol{\beta}^*)]_{S_1} \right\|_2 = \left\| [\widehat{\mathbf{Q}}\boldsymbol{\beta}^* - \mathbf{Q}^*\boldsymbol{\beta}^* + \mathbf{b}^* - \widehat{\mathbf{b}}]_{S_1} \right\|_2 \leq \left\| [\widehat{\mathbf{Q}} - \mathbf{Q}^*]_{S_1 S_1} \right\|_2 \cdot \left\| [\boldsymbol{\beta}^*]_{S_1} \right\|_2 + \left\| [\mathbf{b}^* - \widehat{\mathbf{b}}]_{S_1} \right\|_2$$

$$\leq \sqrt{5}\pi M\sqrt{\frac{\log s_1}{n}} + 4\sqrt{3}\pi\sqrt{\frac{s_1}{n}}$$

holds with probability at least $1 - 2/s_1 - 1/s_1 = 1 - 3/s_1$, where the first term in the last inequality is due to (B.3) and $\|\boldsymbol{\beta}^*\|_2 \leq \|\boldsymbol{\beta}^*\|_1 \leq M$ by Assumption 4.2, and the second term is from Lemma C.4. Thus, we obtain

$$\sum_{i \in S_1} |\left( \nabla\mathcal{L}(\boldsymbol{\beta}^*) + \nabla\mathcal{H}_\lambda(\boldsymbol{\beta}^*) + \lambda\mathbf{z}^* \right)_i| \cdot |(\boldsymbol{\beta}^* - \widehat{\boldsymbol{\beta}})_i| \leq 4\sqrt{3}\pi M\sqrt{\frac{s_1}{n}} \cdot \|\boldsymbol{\beta}^* - \widehat{\boldsymbol{\beta}}\|_2. \tag{A.29}$$

**Case 3:** For $i \in S_2$, we have $|\beta_i^*| \leq \nu$. By condition (d) in Assumption 4.3, we have

$$\max_{i \in S_2} \left| \left( \nabla \mathcal{H}_\lambda(\boldsymbol{\beta}^*) \right)_i \right| \leq \max_{i \in S_2} |h_\lambda'(\beta_i^*)| \leq \max_{1 \leq i \leq d^2} |h_\lambda'(\beta_i^*)| \leq \lambda.$$

Since $\mathbf{z}^* \in \partial \|\boldsymbol{\beta}^*\|_1$, we have $|z_i^*| \leq 1$. Therefore, for $i \in S_2$, the following results hold

$$\left| \left( \nabla \mathcal{L}(\boldsymbol{\beta}^*) + \nabla \mathcal{H}_\lambda(\boldsymbol{\beta}^*) + \lambda \mathbf{z}^* \right)_i \right| \leq \left| \left( \nabla \mathcal{L}(\boldsymbol{\beta}^*) \right)_i \right| + \left| \left( \nabla \mathcal{H}_\lambda(\boldsymbol{\beta}^*) \right)_i \right| + \lambda |(\mathbf{z}^*)_i|$$
$$\leq \left| \left( \nabla \mathcal{L}(\boldsymbol{\beta}^*) \right)_i \right| + 2\lambda.$$

Again, by Lemma B.4 $\|\nabla \mathcal{L}(\boldsymbol{\beta}^*)\|_\infty \leq \lambda/2$ holds with probability at least $1 - 3/d$, we obtain

$$\sum_{i \in S_2} \left| \left( \nabla \mathcal{L}(\boldsymbol{\beta}^*) + \nabla \mathcal{H}_\lambda(\boldsymbol{\beta}^*) + \lambda \mathbf{z}^* \right)_i \right| \cdot |(\boldsymbol{\beta}^* - \widehat{\boldsymbol{\beta}})_i| \leq \left( \|\nabla \mathcal{L}(\boldsymbol{\beta}^*)\|_\infty + 2\lambda \right) \sum_{i \in S_2} |(\boldsymbol{\beta}^* - \widehat{\boldsymbol{\beta}})_i|$$
$$\leq \frac{5}{2}\lambda \sum_{i \in S_2} |(\boldsymbol{\beta}^* - \widehat{\boldsymbol{\beta}})_i|.$$

Hence we have

$$\sum_{i \in S_2} \left| \left( \nabla \mathcal{L}(\boldsymbol{\beta}^*) + \nabla \mathcal{H}_\lambda(\boldsymbol{\beta}^*) + \lambda \mathbf{z}^* \right)_i \right| \cdot |(\boldsymbol{\beta}^* - \widehat{\boldsymbol{\beta}})_i| \leq \frac{5}{2}\lambda\sqrt{s_2}\|(\boldsymbol{\beta}^* - \widehat{\boldsymbol{\beta}})_{S_2}\|_2 \leq \frac{5}{2}\lambda\sqrt{s_2}\|\boldsymbol{\beta}^* - \widehat{\boldsymbol{\beta}}\|_2.$$
$$\text{(A.30)}$$

Adding up (A.28) (A.29) and (A.30), and substituting the right term in (A.27), we obtain

$$\|\widehat{\boldsymbol{\beta}} - \boldsymbol{\beta}^*\|_2 \leq \frac{16\sqrt{3}\pi M}{\kappa_1 \kappa_2}\sqrt{\frac{s_1}{n}} + \frac{10\sqrt{s_2}\lambda}{\kappa_1 \kappa_2} \leq \frac{16\sqrt{3}\pi M}{\kappa_1 \kappa_2}\sqrt{\frac{s_1}{n}} + \frac{10\pi MC}{\kappa_1 \kappa_2}\sqrt{\frac{s_2 \log d}{n}} \quad \text{(A.31)}$$

holds with probability at least $1 - 3/d - 3/s_1 \geq 1 - 6/s_1$. $\qquad\qquad\square$

# B  Lemmas in the Proof of Main Theorems

**Lemma B.1.** Under Assumptions 4.1, the loss function $\mathcal{L}(\boldsymbol{\beta}) = 1/2\boldsymbol{\beta}^\top \widehat{\mathbf{Q}}\boldsymbol{\beta} - \boldsymbol{\beta}^\top \widehat{\mathbf{b}}$ is strongly convex with constant $\kappa_1 \kappa_2 / 2$.

*Proof of Lemma B.1.* Note that $\nabla \mathcal{L}(\boldsymbol{\beta}) = \widehat{\mathbf{Q}}\boldsymbol{\beta} - \widehat{\mathbf{b}}$, we have

$$\langle \nabla \mathcal{L}(\boldsymbol{\beta}) - \nabla \mathcal{L}(\boldsymbol{\beta}'), \boldsymbol{\beta} - \boldsymbol{\beta}' \rangle = \left( \boldsymbol{\beta} - \boldsymbol{\beta}' \right)^\top \widehat{\mathbf{Q}} \left( \boldsymbol{\beta} - \boldsymbol{\beta}' \right).$$

Then we get

$$\min_{\boldsymbol{\beta}, \boldsymbol{\beta}' \in \text{vec}(S)} \left( \boldsymbol{\beta} - \boldsymbol{\beta}' \right)^\top \widehat{\mathbf{Q}} \left( \boldsymbol{\beta} - \boldsymbol{\beta}' \right) = \min_{\boldsymbol{\beta}, \boldsymbol{\beta}' \in \text{vec}(S)} \left( \boldsymbol{\beta} - \boldsymbol{\beta}' \right)^\top \left( \widehat{\mathbf{Q}} - \mathbf{Q}^* + \mathbf{Q}^* \right) \left( \boldsymbol{\beta} - \boldsymbol{\beta}' \right)$$
$$\geq \lambda_{\min}(\mathbf{Q}^*)\|\boldsymbol{\beta} - \boldsymbol{\beta}'\|_2^2 - \max_{\boldsymbol{\beta}, \boldsymbol{\beta}' \in \text{vec}(S)} \left( \boldsymbol{\beta} - \boldsymbol{\beta}' \right)^\top \left( \widehat{\mathbf{Q}} - \mathbf{Q}^* \right) \left( \boldsymbol{\beta} - \boldsymbol{\beta}' \right)$$
$$\geq \lambda_{\min}(\mathbf{Q}^*)\|\boldsymbol{\beta} - \boldsymbol{\beta}'\|_2^2 - \|\widehat{\mathbf{Q}} - \mathbf{Q}^*\|_2 \cdot \|\boldsymbol{\beta} - \boldsymbol{\beta}'\|_2^2.$$

By Assumption 4.1, we have $\lambda_{\min}(\mathbf{Q}^*) = \lambda_{\min}(\boldsymbol{\Sigma}_X^*)\lambda_{\min}(\boldsymbol{\Sigma}_Y^*) = \kappa_1 \kappa_2$. For the second term, we have

$$\|\widehat{\mathbf{Q}} - \mathbf{Q}^*\|_2 = \|\widehat{\boldsymbol{\Sigma}}_X \otimes \widehat{\boldsymbol{\Sigma}}_Y - \boldsymbol{\Sigma}_X^* \otimes \widehat{\boldsymbol{\Sigma}}_Y + \boldsymbol{\Sigma}_X^* \otimes \widehat{\boldsymbol{\Sigma}}_Y + \boldsymbol{\Sigma}_X^* \otimes \boldsymbol{\Sigma}_Y^*\|_2$$
$$\leq \|\widehat{\boldsymbol{\Sigma}}_X - \boldsymbol{\Sigma}_X^*\|_2 \cdot \|\widehat{\boldsymbol{\Sigma}}_Y - \boldsymbol{\Sigma}_Y^*\|_2 + \|\boldsymbol{\Sigma}_X^*\|_2 \cdot \|\widehat{\boldsymbol{\Sigma}}_Y - \boldsymbol{\Sigma}_Y^*\|_2 + \|\boldsymbol{\Sigma}_Y^*\|_2 \cdot \|\widehat{\boldsymbol{\Sigma}}_X - \boldsymbol{\Sigma}_X^*\|_2$$
$$\leq \|\widehat{\boldsymbol{\Sigma}}_X - \boldsymbol{\Sigma}_X^*\|_2 \cdot \|\widehat{\boldsymbol{\Sigma}}_Y - \boldsymbol{\Sigma}_Y^*\|_2 + \frac{1}{\kappa_1} \cdot \|\widehat{\boldsymbol{\Sigma}}_Y - \boldsymbol{\Sigma}_Y^*\|_2 + \frac{1}{\kappa_2} \cdot \|\widehat{\boldsymbol{\Sigma}}_X - \boldsymbol{\Sigma}_X^*\|_2,$$

where the second inequality is due to Assumption 4.1. By Lemma C.1, we have

$$\|\widehat{\boldsymbol{\Sigma}}_X - \boldsymbol{\Sigma}_X^*\|_{\infty,\infty} \leq 3\pi\sqrt{\frac{\log d}{n}}, \qquad \|\widehat{\boldsymbol{\Sigma}}_Y - \boldsymbol{\Sigma}_Y^*\|_{\infty,\infty} \leq 3\pi\sqrt{\frac{\log d}{n}}$$

with probability at least $1 - d^{-5/2}$. Therefore, we have

$$\|(\widehat{\mathbf{Q}} - \mathbf{Q}^*)_S\|_2 \leq 3\pi\sqrt{\frac{s\log d}{n}}\left(\frac{1}{\kappa_1} + \frac{1}{\kappa_2}\right) + \frac{9\pi^2 s\log d}{n}$$

with high probability. When $n$ is sufficient large, we have

$$\min_{\boldsymbol{\beta},\boldsymbol{\beta}'\in C}\left(\boldsymbol{\beta} - \boldsymbol{\beta}'\right)^\top\widehat{\mathbf{Q}}(\boldsymbol{\beta} - \boldsymbol{\beta}') \geq \frac{\kappa_1\kappa_2}{2}\|\boldsymbol{\beta} - \boldsymbol{\beta}'\|_2^2.$$

This immediately implies $\mathcal{L}(\boldsymbol{\beta})$ is restrictively strongly convex with constant $\kappa_1\kappa_2/2$. $\qquad\square$

Lemma (B.1) shows that, with high probability, $\mathcal{L}(\boldsymbol{\beta})$ is a strongly convex function with modulus $\rho = \kappa_1\kappa_2/2 > 0$. In (A.5) we defined $\widetilde{\mathcal{L}}_\lambda(\boldsymbol{\beta}) = \mathcal{L}(\boldsymbol{\beta}) + \mathcal{H}_\lambda(\boldsymbol{\beta})$, where $\mathcal{L}(\boldsymbol{\beta}) = 1/2\boldsymbol{\beta}^\top\widehat{\mathbf{Q}}\boldsymbol{\beta} - \boldsymbol{\beta}^\top\boldsymbol{b}$, $\mathcal{H}_\lambda(\boldsymbol{\beta}) = \sum_{i=1}^{d-1}h_\lambda(\beta_i) = \mathcal{G}_\lambda(\boldsymbol{\beta}) - \lambda\|\boldsymbol{\beta}\|_1$. We now show that, with high probability, $\widetilde{\mathcal{L}}_\lambda(\boldsymbol{\beta})$ is strongly convex.

**Lemma B.2** (Restricted Strongly Convex). Let $S = \text{supp}(\boldsymbol{\beta}^*)$. Given $n \geq C_1 s\log d$ and appropriate parameter in nonconvex penalty $\mathcal{G}_\lambda(\boldsymbol{\beta})$, $\widetilde{\mathcal{L}}_\lambda(\boldsymbol{\beta}')$ is strongly convex.

$$\widetilde{\mathcal{L}}_\lambda(\boldsymbol{\beta}') \geq \widetilde{\mathcal{L}}_\lambda(\boldsymbol{\beta}^*) + \langle\nabla\widetilde{\mathcal{L}}_\lambda(\boldsymbol{\beta}^*),\boldsymbol{\beta}' - \boldsymbol{\beta}^*\rangle + \frac{\rho - \zeta_-}{2}\|\boldsymbol{\beta}' - \boldsymbol{\beta}^*\|_2^2,$$

holds with probability at least $1 - C'\exp(-Cn)$.

*Proof.* Recall that $\mathcal{H}_\lambda(\boldsymbol{\beta})$ is the concave part of $\widetilde{\mathcal{L}}_\lambda(\boldsymbol{\beta})$, which implies $-\mathcal{H}_\lambda(\boldsymbol{\beta})$ is convex. Meanwhile, recall that $\mathcal{H}_\lambda(\boldsymbol{\beta}) = \sum_{i=1}^d h_\lambda(\beta_i)$, where $h_\lambda(\beta_i) + \zeta_-/2\beta_i^2$ is convex by Assumption 4.3. Hence we have

$$h_\lambda(\beta_i') + \frac{\zeta_-}{2}\beta_i'^2 \geq h_\lambda(\beta_i^*) + \frac{\zeta_-}{2}\beta_i^{*2} + \left(h_\lambda'(\beta_i^*) + \zeta_-\beta_i^*\right)(\beta_i' - \beta_i^*),$$

and

$$\mathcal{H}_\lambda(\boldsymbol{\beta}') + \frac{\zeta_-}{2}\|\boldsymbol{\beta}'\|_2^2 \geq \mathcal{H}_\lambda(\boldsymbol{\beta}^*) + \frac{\zeta_-}{2}\|\boldsymbol{\beta}^*\|_2^2 + \langle\nabla\mathcal{H}_\lambda(\boldsymbol{\beta}^*) + \zeta_-\boldsymbol{\beta}^*,\boldsymbol{\beta}' - \boldsymbol{\beta}^*\rangle.$$

This immediately implies

$$\mathcal{H}_\lambda(\boldsymbol{\beta}') \geq \mathcal{H}_\lambda(\boldsymbol{\beta}^*) + \langle\nabla\mathcal{H}_\lambda(\boldsymbol{\beta}^*),\boldsymbol{\beta}' - \boldsymbol{\beta}^*\rangle - \frac{\zeta_-}{2}\|\boldsymbol{\beta}' - \boldsymbol{\beta}^*\|_2^2. \tag{B.1}$$

Recall that by Lemma B.1, provided suitable condition, $\mathcal{L}(\boldsymbol{\beta}')$ is (w.h.p.) strongly convex. with modulus $\rho = \kappa_1\kappa_2/2$, we have

$$\mathcal{L}(\boldsymbol{\beta}') \geq \mathcal{L}(\boldsymbol{\beta}^*) + \langle\nabla\mathcal{L}(\boldsymbol{\beta}^*),\boldsymbol{\beta}' - \boldsymbol{\beta}^*\rangle + \frac{\rho}{2}\|\boldsymbol{\beta}' - \boldsymbol{\beta}^*\|_2^2. \tag{B.2}$$

By the definition of $\widetilde{\mathcal{L}}_\lambda(\boldsymbol{\beta}) = \mathcal{L}(\boldsymbol{\beta}) + \mathcal{H}_\lambda(\boldsymbol{\beta})$, adding (B.1) and (B.2) together, we obtain

$$\widetilde{\mathcal{L}}_\lambda(\boldsymbol{\beta}') \geq \widetilde{\mathcal{L}}_\lambda(\boldsymbol{\beta}^*) + \langle\nabla\widetilde{\mathcal{L}}_\lambda(\boldsymbol{\beta}^*),\boldsymbol{\beta}' - \boldsymbol{\beta}^*\rangle + \frac{\rho - \zeta_-}{2}\|\boldsymbol{\beta}' - \boldsymbol{\beta}^*\|_2^2,$$

holds with probability at least $1 - C'\exp(-Cn)$. Here, $\rho = \kappa_1\kappa_2/2$ and $\zeta_-$ is depended on the nonconvex penalty. For example, in MCP penalty $\zeta_- = 1/b$. When $\rho = \kappa_1\kappa_2/2 > 1/b$, the above equation leads to strongly convexity of $\widetilde{\mathcal{L}}_\lambda(\boldsymbol{\beta})$, w.h.p. in the cone, provided suitable condition on $n$. $\qquad\square$

**Lemma B.3.** Under Assumption 4.1, the oracle estimator $\widehat{\boldsymbol{\beta}}_O$ in (A.6) satisfies

$$\left\|\widehat{\boldsymbol{\beta}}_O - \boldsymbol{\beta}^*\right\|_\infty \leq 6\pi\theta_X\theta_Y\sqrt{\frac{\log s}{n}} + 2\sqrt{10}\pi\theta_X^2\theta_Y^2\sigma_X\sigma_Y M\sqrt{\frac{\log s}{n}},$$

with probability at least $1 - 3/s$.

*Proof.* By definition in (A.6),we have

$$\widehat{\boldsymbol{\beta}}_O = \underset{\mathrm{supp}(\boldsymbol{\beta}) \subseteq \mathrm{vec}(S)}{\mathrm{argmin}} \frac{1}{2} \boldsymbol{\beta}^\top \widehat{\mathbf{Q}} \boldsymbol{\beta} - \widehat{\mathbf{b}}^\top \boldsymbol{\beta}.$$

By definition we have $\widehat{\mathbf{Q}} = \widehat{\boldsymbol{\Sigma}}_X \otimes \widehat{\boldsymbol{\Sigma}}_Y \in \mathbb{R}^{d^2 \times d^2}$. For any $j, k, p, q = 1, \dots, d$, we use $\widehat{\mathbf{Q}}_{(j,k,p,q)}$ to denote the entry in $\widehat{\mathbf{Q}}$ that is obtained from the product of the $(j, k)$-th entry in $\widehat{\boldsymbol{\Sigma}}_X$ and the $(p, q)$-th entry in $\widehat{\boldsymbol{\Sigma}}_Y$. Specifically, we have

$$\widehat{\mathbf{Q}}_{(j,k,p,q)} = \widehat{\Sigma}_X^{jk} \widehat{\Sigma}_Y^{pq} = \sin\left(\frac{\pi}{2}\widehat{\tau}_{jk}^X\right) \sin\left(\frac{\pi}{2}\widehat{\tau}_{pq}^Y\right)$$

$$= \frac{1}{2}\cos\left(\frac{\pi}{2}\left(\widehat{\tau}_{jk}^X - \widehat{\tau}_{pq}^Y\right)\right) - \frac{1}{2}\cos\left(\frac{\pi}{2}\left(\widehat{\tau}_{jk}^X + \widehat{\tau}_{pq}^Y\right)\right).$$

Furthermore, we define $\widehat{\mu}_{jk;pq} = \widehat{\tau}_{jk}^X - \widehat{\tau}_{pq}^Y$ and $\widehat{\mu}'_{jk;pq} = \widehat{\tau}_{jk}^X + \widehat{\tau}_{pq}^Y$. All the notations above can be easily extended to $\mathbf{Q}^*$. Then we have

$$\widehat{\mathbf{Q}}_{(j,k,p,q)} - \mathbf{Q}^*_{(j,k,p,q)} = \frac{1}{2}\left(\cos\left(\frac{\pi}{2}\widehat{\mu}_{jk;pq}\right) - \cos\left(\frac{\pi}{2}\mu^*_{jk;pq}\right)\right)$$

$$+ \frac{1}{2}\left(\cos\left(\frac{\pi}{2}\widehat{\mu}'_{jk;pq}\right) - \cos\left(\frac{\pi}{2}\mu'^*_{jk;pq}\right)\right).$$

We only need to bound the first term, and the second term is very similar. Note that

$$\cos\left(\frac{\pi}{2}\widehat{\mu}_{jk;pq}\right) - \cos\left(\frac{\pi}{2}\mu^*_{jk;pq}\right) = -\frac{\pi}{2}\sin\left(\frac{\pi}{2}\widetilde{\mu}_{jk;pq}\right)\left(\widehat{\mu}_{jk;pq} - \mu^*_{jk;pq}\right),$$

where $\widetilde{\mu}_{jk;pq}$ lies between $\widehat{\mu}_{jk;pq}$ and $\mu^*_{jk;pq}$. To bound $\widehat{\mu}_{jk;pq} - \mu^*_{jk;pq}$, note that $\widehat{\tau}_{jk}, \widehat{\tau}_{pq}$ are sub-Gaussian random variables and $|\widehat{\tau}_{jk}^X|, |\widehat{\tau}_{pq}^Y| \leq 1$. Thus $|\widehat{\mu}_{jk;pq}| \leq 2$ and $\widehat{\mu}_{jk;pq}$ is also sub-Gaussian. In addition, we have $\mathbb{E}\left(\widehat{\mu}_{jk;pq}\right) = \mathbb{E}(\widehat{\tau}_{jk}^X) - \mathbb{E}(\widehat{\tau}_{pq}^Y) = \mu^*_{jk;pq}$. Then by Hoeffding's inequality for U-statistics and applying union bound, we get

$$\mathbb{P}\left(\sup_{j,k,p,q} |\widehat{\mu}_{jk;pq} - \mu^*_{jk;pq}| > t\right) \leq 2d^4 e^{-\frac{nt^2}{4}}.$$

Take $t = \sqrt{20 \log d / n}$, we have that

$$\sup_{j,k,p,q} \left|\cos\left(\frac{\pi}{2}\widehat{\mu}_{jk;pq}\right) - \cos\left(\frac{\pi}{2}\mu^*_{jk;pq}\right)\right| \leq \frac{\pi}{2}\sup_{j,k,p,q} |\widehat{\mu}_{jk;pq} - \mu^*_{jk;pq}| \leq \frac{\pi}{2}\sqrt{\frac{20 \log d}{n}}$$

holds with probability at least $1 - 2/d$. It follows that

$$\left\|\widehat{\mathbf{Q}} - \mathbf{Q}\right\|_{\infty,\infty} = \sup_{j,k,p,q} \left|\widehat{\mathbf{Q}}_{(j,k,p,q)} - \mathbf{Q}^*_{(j,k,p,q)}\right| \leq \sqrt{5}\pi\sqrt{\frac{\log d}{n}} \qquad (B.3)$$

holds with probability at least $1 - 2/d$. We have the closed form solution of $\widehat{\boldsymbol{\beta}}_O$ as $\widehat{\boldsymbol{\beta}}_O = \widehat{\mathbf{Q}}_{SS}^{-1}\widehat{\mathbf{b}}$. Then we have

$$\left\|\widehat{\boldsymbol{\beta}}_O - \boldsymbol{\beta}^*\right\|_\infty = \left\|\widehat{\mathbf{Q}}_{SS}^{-1}\widehat{\mathbf{b}} - \mathbf{Q}^{*-1}\mathbf{b}^*\right\|_\infty = \left\|\widehat{\mathbf{Q}}_{SS}^{-1}\widehat{\mathbf{b}} - \widehat{\mathbf{Q}}_{SS}^{-1}\mathbf{b}^* + \widehat{\mathbf{Q}}_{SS}^{-1}\mathbf{b}^* - \mathbf{Q}^{*-1}\mathbf{b}^*\right\|_\infty$$

$$\leq \underbrace{\left\|\widehat{\mathbf{Q}}_{SS}^{-1}\right\|_1 \cdot \left\|[\widehat{\mathbf{b}} - \mathbf{b}^*]_S\right\|_\infty}_{(i)} + \underbrace{\left\|\widehat{\mathbf{Q}}_{SS}^{-1} - \mathbf{Q}_{SS}^{*-1}\right\|_{\infty,\infty} \cdot \|\mathbf{b}_S^*\|_1}_{(ii)}.$$

$$(B.4)$$

For term (ii), we have

$$\left\|\widehat{\mathbf{Q}}_{SS}^{-1} - \mathbf{Q}_{SS}^{*-1}\right\|_{\infty,\infty} = \left\|\mathbf{Q}_{SS}^{*-1}\left(\widehat{\mathbf{Q}}_{SS} - \mathbf{Q}_{SS}^*\right)\widehat{\mathbf{Q}}_{SS}^{-1}\right\|_{\infty,\infty}$$

$$\leq \left\|\mathbf{Q}_{SS}^{*-1}\right\|_1 \cdot \left\|\widehat{\mathbf{Q}}_{SS} - \mathbf{Q}_{SS}^*\right\|_{\infty,\infty} \cdot \left\|\widehat{\mathbf{Q}}_{SS}^{-1}\right\|_1.$$

By Assumption we have $\left\|\mathbf{Q}^{*-1}\right\|_1 = \left\|\boldsymbol{\Sigma}_X^{*-1}\right\|_1 \cdot \left\|\boldsymbol{\Sigma}_X^{*-1}\right\|_1 \leq \theta_X\theta_Y$. When $n$ is sufficient large, we have $\left\|\widehat{\mathbf{Q}}^{-1}\right\|_1 \leq 2\left\|\mathbf{Q}^{*-1}\right\|_1$ by concentration. By (B.3) we have with probability at least $1 - 2/s$ that

$$\left\|\widehat{\mathbf{Q}}_{SS}^{-1} - \mathbf{Q}_{SS}^{*-1}\right\|_{\infty,\infty} \cdot \|\mathbf{b}^*\|_1 \leq 2\sqrt{10}\pi\theta_X^2\theta_Y^2\sigma_X\sigma_Y M\sqrt{\frac{\log s}{n}}, \qquad (B.5)$$

where we used the fact that $\|\boldsymbol{\Sigma}_X^* - \boldsymbol{\Sigma}_Y^*\|_{1,1} \le \sigma_X \sigma_Y M$ by Assumption 4.2. For term (i), we have with probability at least $1 - s^{-2.5}$

$$\left\|[\widehat{\mathbf{b}} - \mathbf{b}^*]_S\right\|_\infty \le \left\|[\widehat{\boldsymbol{\Sigma}}_X - \boldsymbol{\Sigma}_X^*]_S\right\|_{\infty,\infty} + \left\|[\widehat{\boldsymbol{\Sigma}}_Y - \boldsymbol{\Sigma}_Y^*]_S\right\|_{\infty,\infty} \le 6\pi\sqrt{\frac{\log s}{n}}, \tag{B.6}$$

where the second inequality is due to Lemma C.1. Therefore, submitting (B.5) and (B.6) into (B.4), we obtain

$$\left\|\widehat{\boldsymbol{\beta}}_O - \boldsymbol{\beta}^*\right\|_\infty \le 6\pi\theta_X\theta_Y\sqrt{\frac{\log s}{n}} + 2\sqrt{10}\pi\theta_X^2\theta_Y^2\sigma_X\sigma_Y M\sqrt{\frac{\log s}{n}},$$

which holds with probability at least $1 - 2/s - 1/s^{2.5} \ge 1 - 3/s$.

$\square$

**Lemma B.4.** We have with probability at least $1 - 3/d$ that

$$\|\mathcal{L}(\boldsymbol{\beta}^*)\|_\infty \le CM\sqrt{\frac{\log d}{n}},$$

where $C$ is an absolute constants.

*Proof of Lemma B.4.* Since $\nabla\mathcal{L}(\boldsymbol{\beta}) = \widehat{\mathbf{Q}}\boldsymbol{\beta} - \widehat{\mathbf{b}}$ and note that $\mathbf{Q}^*\boldsymbol{\beta}^* = \mathbf{b}^*$, we have

$$\begin{aligned}
\left\|\nabla\mathcal{L}(\boldsymbol{\beta}^*)\right\|_\infty &= \left\|\widehat{\mathbf{Q}}\boldsymbol{\beta}^* - \mathbf{Q}^*\boldsymbol{\beta}^* + \mathbf{b}^* - \widehat{\mathbf{b}}\right\|_\infty \\
&\le \left\|\widehat{\mathbf{Q}} - \mathbf{Q}^*\right\|_{\infty,\infty} \cdot \left\|\boldsymbol{\beta}^*\right\|_1 + \left\|\mathbf{b}^* - \widehat{\mathbf{b}}\right\|_\infty \\
&\le \sqrt{5}\pi M\sqrt{\frac{\log d}{n}} + 6\pi\sqrt{\frac{\log d}{n}}
\end{aligned}$$

holds with probability at least $1 - 2/d - 1/d^{2.5} \ge 1 - 3/d$, where the first term in the last inequality is due to (B.3) and $\|\boldsymbol{\beta}^*\|_1 \le M$ by Assumption 4.2, and the second term is due to Lemma C.1. $\square$

# C   Auxiliary Lemmas

**Lemma C.1.** [20] Given $X_1, X_2, \ldots, X_n$ are i.i.d. random vectors following $TE_d(\boldsymbol{\Sigma}^*, \xi; f_1, f_2, \ldots, f_d)$ and letting $\widehat{\boldsymbol{\Sigma}}$ be the Kendall tau correlation matrix, we have that

$$\|\widehat{\boldsymbol{\Sigma}} - \boldsymbol{\Sigma}^*\|_{\infty,\infty} \le 3\pi\sqrt{\frac{\log d}{n}}$$

holds with probability at least $1 - d^{-5/2}$.

To prove the spectral norm error of $\widehat{\boldsymbol{\Sigma}}$, we first introduce the following bound for $\widehat{\mathbf{T}}$, where $\widehat{T}_{jk} = \widehat{\tau}_{jk}$.

**Lemma C.2.** Suppose $\delta \in (0,1)$ satisfy $\log(1/\delta) + d\log(9) \le n$. Then with probability $1 - \delta$ it holds that

$$\sup_{\|\mathbf{x}\|_2 \le 1} \left|\mathbf{x}^\top\left(\widehat{\mathbf{T}} - \mathbf{T}^*\right)\mathbf{x}\right| \le 4\sqrt{C}\sqrt{\frac{\log(1/\delta) + d\log(9)}{n}},$$

where $C$ is an absolute constant.

*Proof of Lemma C.2.* Let $\theta = 4\sqrt{C}\sqrt{[\log(1/\delta) + s\log(12)]/n}$. For any fixed $\mathbf{x}$ with $\|\mathbf{x}\|_2 \le 1$ and any $0 < t$, by Markov's inequality

$$\mathbb{P}\left(\mathbf{x}^\top\left(\widehat{\mathbf{T}} - \mathbf{T}^*\right)\mathbf{x} > \theta\right) \le \mathbb{E}\left[\exp\left(t \cdot \mathbf{x}^\top\left(\widehat{\mathbf{T}} - \mathbf{T}^*\right)\mathbf{x} - t\theta\right)\right]. \tag{C.1}$$

By Lemma D.1, we have

$$\mathbb{E}\left[\exp\left(t \cdot \mathbf{x}^\top(\widehat{\mathbf{T}} - \mathbf{T}^*)\mathbf{x}\right)\right] \le \exp\left(\frac{8Ct^2}{n}\right).$$

Submitting the above inequality into (C.1) and setting $t = n\theta/(16C)$, we obtain

$$\mathbb{P}\Big(\mathbf{x}^\top\big(\widehat{\mathbf{T}} - \mathbf{T}^*\big)\mathbf{x} > \theta\Big) \leq \exp\Big(-\frac{n\theta^2}{16C}\Big). \tag{C.2}$$

Thus the error bound for $\widehat{\mathbf{T}}$ in spectral norm satisfies

$$\begin{aligned}
\mathbb{P}\big(\|\widehat{\mathbf{T}} - \mathbf{T}^*\|_2 > \theta\big) &= \mathbb{P}\bigg(\sup_{\|\mathbf{x}\|_2 = 1} \mathbf{x}^\top\big(\widehat{\mathbf{T}} - \mathbf{T}^*\big)\mathbf{x} > \theta\bigg) \\
&\leq \mathbb{P}\bigg((1 - 2\epsilon)^{-1} \sup_{\mathbf{x} \in \mathcal{N}_\epsilon} \mathbf{x}^\top\big(\widehat{\mathbf{T}} - \mathbf{T}^*\big)\mathbf{x} > \theta\bigg) \\
&\leq (1 + 2/\epsilon)^d \mathbb{P}\big(\mathbf{x}^\top\big(\widehat{\mathbf{T}} - \mathbf{T}^*\big)\mathbf{x} > (1 - 2\epsilon)\theta\big),
\end{aligned}$$

where the first inequality is due to Lemma D.3 and the second one due to Lemma D.2. Take $\epsilon = 1/4$ we obtain

$$\mathbb{P}\big(\|\widehat{\mathbf{T}} - \mathbf{T}^*\|_2 > \theta\big) \leq 9^d \mathbb{P}\big(\mathbf{x}^\top\big(\widehat{\mathbf{T}} - \mathbf{T}^*\big)\mathbf{x} > \frac{\theta}{2}\big) \leq 9^d \exp\Big(-\log(1/\delta) - d\log 9\Big) = \delta.$$

where in the second inequality we used (C.2) and the definition of $\theta$. This completes the proof. $\quad\square$

Next, we relate $\widehat{\boldsymbol{\Sigma}}$ to $\widehat{\mathbf{T}}$. We have the following bound on the error of covariance estimator $\widehat{\boldsymbol{\Sigma}}$:

**Lemma C.3.** Assume that $X_1, \ldots, X_n$ are i.i.d. random vectors following $TE_d(\boldsymbol{\Sigma}^*, \xi; f_1, f_2, \ldots, f_d)$ and letting $\widehat{\boldsymbol{\Sigma}}$ be the Kendall tau correlation matrix, we have

$$\|\widehat{\boldsymbol{\Sigma}} - \boldsymbol{\Sigma}^*\|_2 \leq 2\pi^2 \frac{d\log d}{n} + 8\pi\sqrt{C}\sqrt{\frac{\log d + d\log(9)}{n}}$$

holds with probability at least $1 - 1/d - 2/d^2$.

*Proof of Lemma C.3.* By definition in Section 3.2, we have $\widehat{\boldsymbol{\Sigma}} = \cos\big(\pi/2\widehat{\mathbf{T}}\big)$, where the $\cos(\cdot)$ function is elementwise. By Taylor's theorem,

$$\widehat{\boldsymbol{\Sigma}} = \boldsymbol{\Sigma}^* + \frac{\pi}{2}\cos\Big(\frac{\pi}{2}\mathbf{T}^*\Big) \circ \big(\widehat{\mathbf{T}} - \mathbf{T}^*\big) - \frac{\pi^2}{8}\sin\Big(\frac{\pi}{2}\widetilde{\mathbf{T}}\Big) \circ \big(\widehat{\mathbf{T}} - \mathbf{T}^*\big) \circ \big(\widehat{\mathbf{T}} - \mathbf{T}^*\big),$$

where $\widetilde{\mathbf{T}}$ has entries $\widetilde{\tau}_{jk}$ which lies between $\widehat{\tau}_{jk}$ and $\tau_{jk}^*$ for all $j, k = 1, \cdot, d$. Here $\circ$ is the Hadamard (elementwise) product for matrices. For any $\mathbf{x} \in \mathbb{R}^d$ and $\|\mathbf{x}\|_2 \leq 1$,

$$\big|\mathbf{x}^\top\big(\widehat{\boldsymbol{\Sigma}} - \boldsymbol{\Sigma}\big)\mathbf{x}\big| \leq \underbrace{\frac{\pi}{2}\Big|\mathbf{x}^\top\Big[\cos\Big(\frac{\pi}{2}\mathbf{T}^*\Big) \circ \big(\widehat{\mathbf{T}} - \mathbf{T}^*\big)\Big]\mathbf{x}\Big|}_{(i)} + \underbrace{\frac{\pi^2}{8}\Big|\mathbf{x}^\top\Big[\sin\Big(\frac{\pi}{2}\widetilde{\mathbf{T}}\Big) \circ \big(\widehat{\mathbf{T}} - \mathbf{T}^*\big) \circ \big(\widehat{\mathbf{T}} - \mathbf{T}^*\big)\Big]\mathbf{x}\Big|}_{(ii)}. \tag{C.3}$$

We first bound the term (ii). Note that

$$\begin{aligned}
\Big|\mathbf{x}^\top\Big[\sin\Big(\frac{\pi}{2}\widetilde{\mathbf{T}}\Big) \circ \big(\widehat{\mathbf{T}} - \mathbf{T}^*\big) \circ \big(\widehat{\mathbf{T}} - \mathbf{T}^*\big)\Big]\mathbf{x}\Big| &\leq \|\mathbf{x}\|_1^2 \cdot \Big\|\sin\Big(\frac{\pi}{2}\widetilde{\mathbf{T}}\Big) \circ \big(\widehat{\mathbf{T}} - \mathbf{T}^*\big) \circ \big(\widehat{\mathbf{T}} - \mathbf{T}^*\big)\Big\|_{\infty,\infty} \\
&\leq d \cdot \big\|\widehat{\mathbf{T}} - \mathbf{T}^*\big\|_{\infty,\infty}^2,
\end{aligned}$$

where the second inequality holds because $|\sin(\pi/2\widetilde{\tau}_{jk})| \leq 1$, for all $j, k = 1, \cdots, d$, and $\|\mathbf{x}\|_1 \leq \sqrt{d}\|\mathbf{x}\|_2 \leq \sqrt{d}$. Next, we bound term (i) in (C.3). By Lemma D.4, we can express $\cos(\pi/2\mathbf{T}^*)$ as a convex combination,

$$\cos\Big(\frac{\pi}{2}\mathbf{T}^*\Big) = \sum_{i=1}^\infty a_i \mathbf{u}_i \mathbf{v}_i^\top,$$

where $\mathbf{u}_i, \mathbf{v}_i \in \mathbb{R}^d$ satisfy $\|\mathbf{u}_i\|_\infty, \|\mathbf{v}_i\|_\infty \leq 1$ for all $i \geq 1$, and the non-negative sequence $a_1, a_2, \cdots$ satisfy $\sum_{i=1}^\infty a_i = 4$. Then

$$\left|\mathbf{x}^\top \left[\cos\left(\frac{\pi}{2}\mathbf{T}^*\right) \circ (\widehat{\mathbf{T}} - \mathbf{T}^*)\right]\mathbf{x}\right| \leq \sum_{i=1}^\infty a_i \left|\mathbf{x}^\top \left[\mathbf{u}_i \mathbf{v}_i^\top \circ (\widehat{\mathbf{T}} - \mathbf{T}^*)\right]\mathbf{x}\right|$$

$$= \sum_{i=1}^\infty a_i \left|(\mathbf{x} \circ \mathbf{v})^\top (\widehat{\mathbf{T}} - \mathbf{T}^*)(\mathbf{x} \circ \mathbf{u})\right|$$

$$\leq 4 \sup_{\mathbf{u}_0, \mathbf{v}_0} \left|\mathbf{v}_0^\top (\widehat{\mathbf{T}} - \mathbf{T}^*)\mathbf{u}_0\right|,$$

where $\mathbf{u}_0 = \mathbf{x} \circ \mathbf{u}, \mathbf{v}_0 = \mathbf{x} \circ \mathbf{v}$. Note that $\|\mathbf{u}_0\|_2 \leq \|\mathbf{x}\|_2 \cdot \|\mathbf{u}\|_\infty \leq 1$, similarly $\|\mathbf{v}\|_2 \leq 1$. Let $\widetilde{\mathbf{u}} = (\mathbf{u}_0 + \mathbf{v}_0)/2$ and $\widetilde{\mathbf{v}} = (\mathbf{u}_0 - \mathbf{v}_0)/2$. Observe that we have $\|\widetilde{\mathbf{u}}\|_2 \leq (\|\mathbf{u}_0\|_2 + \|\mathbf{v}_0\|_2)/2 \leq 1$, similarly $\|\widetilde{\mathbf{v}}\|_2 \leq 1$. Therefore,

$$\sup_{\|\mathbf{u}_0\|_2, \|\mathbf{v}_0\|_2 \leq 1} \left|\mathbf{v}_0^\top (\widehat{\mathbf{T}} - \mathbf{T}^*)\mathbf{u}_0\right| = \frac{1}{2} \sup_{\|\widetilde{\mathbf{u}}\|_2, \|\widetilde{\mathbf{v}}\|_2 \leq 1} \left|\widetilde{\mathbf{u}}^\top (\widehat{\mathbf{T}} - \mathbf{T}^*)\widetilde{\mathbf{u}} - \widetilde{\mathbf{v}}^\top (\widehat{\mathbf{T}} - \mathbf{T}^*)\widetilde{\mathbf{v}}\right|$$

$$\leq \sup_{\|\widetilde{\mathbf{u}}\|_2 \leq 1} \left|\widetilde{\mathbf{u}}^\top (\widehat{\mathbf{T}} - \mathbf{T}^*)\widetilde{\mathbf{u}}\right|.$$

Recall the bound in (C.3), we now obtain

$$\sup_{\|\mathbf{x}\|_2 \leq 1} \left|\mathbf{x}^\top (\widehat{\mathbf{\Sigma}} - \mathbf{\Sigma}^*)\mathbf{x}\right| \leq \frac{\pi^2}{8} \cdot d\|\widehat{\mathbf{T}} - \mathbf{T}\|_{\infty,\infty}^2 + 2\pi \sup_{\|\mathbf{x}\|_2 \leq 1} \left|\mathbf{x}^\top (\widehat{\mathbf{T}} - \mathbf{T}^*)\mathbf{x}\right|. \quad \text{(C.4)}$$

Since $\widehat{\tau}_{jk}$ is a U-statistic, and its kernel is a bounded function between $-1$ and $1$ and $\mathbb{E}\widehat{\tau}_{jk} = \tau_{jk}$. Then by Hoeffding's inequality for U-statistics, we obtain

$$\mathbb{P}\left(\sup_{j,k} |\widehat{\tau}_{jk} - \tau_{jk}| > t\right) \leq 2d^2 e^{-\frac{nt^2}{4}}. \quad \text{(C.5)}$$

Choose $t = 4\sqrt{\log d/n}$, we have $\|\widehat{\mathbf{T}} - \mathbf{T}\|_{\infty,\infty} \leq 4\sqrt{\log d/n}$ with probability at least $1 - 2/d^2$. Plugging the bound in Lemma C.2 and (C.5) into (C.4), we obtain that

$$\|\widehat{\mathbf{\Sigma}} - \mathbf{\Sigma}^*\|_2 \leq 2\pi^2 \frac{d\log d}{n} + 8\pi\sqrt{C}\sqrt{\frac{\log d + d\log(9)}{n}}$$

holds with probability at least $1 - 1/d - 2/d^2$, where we set $\delta = 1/d$ in Lemma C.2 and $C$ is an absolute constant. $\qquad\square$

**Lemma C.4.** For vectors $\widehat{\mathbf{b}}, \mathbf{b}^* \in \mathbb{R}^{d^2}$ with entries $\widehat{b}_j = \sin(\widehat{\tau}_j)$, and a index set $S$ with $|S| = s$, we have

$$\left\|[\widehat{\mathbf{b}} - \mathbf{b}^*]_S\right\|_2 \leq 4\sqrt{3}\pi\sqrt{\frac{s}{n}}$$

holds with probability at least $1 - 1/s$.

*Proof of Lemma C.4.* By definition $\widehat{\mathbf{b}} = \text{vec}(\widehat{\mathbf{\Sigma}}_X - \widehat{\mathbf{\Sigma}}_Y)$,

$$\|\widehat{\mathbf{b}} - \mathbf{b}^*\|_2 \leq \|\text{vec}(\widehat{\mathbf{\Sigma}}_X) - \text{vec}(\mathbf{\Sigma}_X^*)\|_2 + \|\text{vec}(\widehat{\mathbf{\Sigma}}_Y) - \text{vec}(\mathbf{\Sigma}_Y^*)\|_2.$$

Denote $\widehat{\mathbf{x}} = \text{vec}(\widehat{\mathbf{\Sigma}}_X), \mathbf{x}^* = \text{vec}(\mathbf{\Sigma}_X^*)$ and $\widehat{\mathbf{y}} = \text{vec}(\widehat{\mathbf{\Sigma}}_Y), \mathbf{y}^* = \text{vec}(\mathbf{\Sigma}_Y^*)$. We only need to bound $\|\widehat{\mathbf{x}} - \mathbf{x}^*\|_2$. By definition in Section 3.2, we have $\widehat{\mathbf{\Sigma}} = \cos(\pi/2\widehat{\mathbf{T}})$ and $\widehat{\mathbf{x}} = \cos(\pi/2\tau)$, where the $\cos(\cdot)$ function is elementwise. Here we use $\tau = \text{vec}(\widehat{\mathbf{T}})$ to denote the vectorized Tau statistic. By Taylor's theorem,

$$\widehat{\mathbf{x}} = \mathbf{x}^* + \frac{\pi}{2}\cos\left(\frac{\pi}{2}\tau^*\right) \circ (\widehat{\tau} - \tau^*) - \frac{\pi^2}{8}\sin\left(\frac{\pi}{2}\widetilde{\tau}\right) \circ (\widehat{\tau} - \tau^*) \circ (\widehat{\tau} - \tau^*),$$

where $\widetilde{\boldsymbol{\tau}}$ has entries $\widetilde{\tau}_j$ which lies between $\widehat{\tau}_j$ and $\tau_j^*$ for all $j = 1, \cdots, d^2$ and $\circ$ is the Hadamard (elementwise) product. For any $\mathbf{u} \in \mathbb{R}^s$ and $\|\mathbf{u}\|_2 \leq 1$,

$$\left|\mathbf{u}^\top (\widehat{\mathbf{x}} - \mathbf{x})_S\right| \leq \underbrace{\frac{\pi}{2}\left|\mathbf{u}^\top\left[\cos\left(\frac{\pi}{2}\boldsymbol{\tau}^*\right) \circ (\widehat{\boldsymbol{\tau}} - \boldsymbol{\tau}^*)\right]_S\right|}_{\text{(i)}} + \underbrace{\frac{\pi^2}{8}\left|\mathbf{u}^\top\left[\sin\left(\frac{\pi}{2}\widetilde{\boldsymbol{\tau}}\right) \circ (\widehat{\boldsymbol{\tau}} - \boldsymbol{\tau}^*) \circ (\widehat{\boldsymbol{\tau}} - \boldsymbol{\tau}^*)\right]_S\right|}_{\text{(ii)}}.$$

$$\text{(C.6)}$$

We first bound the term (i). Note that

$$\left|\mathbf{u}^\top\left[\sin\left(\frac{\pi}{2}\widetilde{\boldsymbol{\tau}}\right) \circ (\widehat{\boldsymbol{\tau}} - \boldsymbol{\tau}^*) \circ (\widehat{\boldsymbol{\tau}} - \boldsymbol{\tau}^*)\right]_S\right| \leq \|\mathbf{u}\|_1 \cdot \left\|\left[\sin\left(\frac{\pi}{2}\widetilde{\boldsymbol{\tau}}\right) \circ (\widehat{\boldsymbol{\tau}} - \boldsymbol{\tau}^*) \circ (\widehat{\boldsymbol{\tau}} - \boldsymbol{\tau}^*)\right]_S\right\|_\infty$$

$$\leq \sqrt{s} \cdot \left\|[\widehat{\mathbf{T}} - \mathbf{T}^*]_S\right\|_{\infty,\infty}^2,$$

where the second inequality holds because $|\sin(\pi/2\widetilde{\tau}_j)| \leq 1$, for all $j \in S$, and $\|\mathbf{u}\|_1 \leq \sqrt{s}\|\mathbf{u}\|_2 \leq \sqrt{s}$. By (C.5), we have $\|[\widehat{\mathbf{T}} - \mathbf{T}^*]_S\|_\infty \leq 4\sqrt{\log s/n}$.

Next, we bound term (ii) in (C.6). Note that

$$\left|\mathbf{u}^\top\left[\cos\left(\frac{\pi}{2}\boldsymbol{\tau}^*\right) \circ (\widehat{\boldsymbol{\tau}} - \boldsymbol{\tau}^*)\right]_S\right| = \left|\left[\mathbf{u} \circ \cos\left(\frac{\pi}{2}\boldsymbol{\tau}_S^*\right)\right]^\top (\widehat{\boldsymbol{\tau}} - \boldsymbol{\tau}^*)_S\right| = |\mathbf{u}_1^\top(\widehat{\boldsymbol{\tau}} - \boldsymbol{\tau}^*)_S|,$$

where $\mathbf{u}_1 \in \mathbb{R}^s$ is a constant vector with $\|\mathbf{u}_1\|_2 \leq 1$. Since $\widehat{\tau}_j$ is a U-statistic, and its kernel is a bounded function between $-1$ and $1$ and $\mathbb{E}\widehat{\tau}_j = \tau_j^*$. Thus $\tau_j - \tau_j^*$ are centered sub-Gaussian random variables and $\|\tau_j - \tau_j^*\|_{\psi_2} \leq 2$. Then by Hoeffding's inequality, we obtain

$$\mathbb{P}\left(|\mathbf{u}_1^\top(\widehat{\boldsymbol{\tau}} - \boldsymbol{\tau}^*)_S| > t\right) \leq e^{-\frac{nt^2}{4}}. \qquad \text{(C.7)}$$

By Lemma D.2 and Lemma D.3

$$\mathbb{P}\left(\sup_{\mathbf{u}_1 \in \mathbb{R}^s, \|\mathbf{u}_1\|_2 \leq 1} |\mathbf{u}_1^\top(\widehat{\boldsymbol{\tau}} - \boldsymbol{\tau}^*)_S| > t\right) \leq \mathbb{P}\left(\sup_{\mathbf{u}_1 \in S_\epsilon} |\mathbf{u}_1^\top(\widehat{\boldsymbol{\tau}} - \boldsymbol{\tau}^*)_S| > (1-\epsilon)^{-1}t\right)$$

$$\leq (1 + 2/\epsilon)^s \mathbb{P}\left(|\mathbf{u}_1^\top(\widehat{\boldsymbol{\tau}} - \boldsymbol{\tau}^*)_S| > (1-\epsilon)^{-1}t\right)$$

$$\leq 5^s \exp\left(-\frac{nt^2}{16}\right),$$

where we set $\epsilon = 1/2$ for the $\epsilon$-net of $s$-sphere. Choose $t = 4\sqrt{3s/n}$ and then we have $\|[\widehat{\boldsymbol{\tau}} - \boldsymbol{\tau}^*]_S\|_2 \leq 4\sqrt{3s/n}$ with probability at least $1 - 1/s$. Therefore, we obtain

$$\|[\widehat{\mathbf{x}} - \mathbf{x}^*]_S\|_2 = \max_{\|\mathbf{u}\|_2 = 1} |\mathbf{u}^\top(\widehat{\mathbf{x}} - \mathbf{x})_S| \leq 2\pi^2\frac{\sqrt{s}\log s}{n} + 2\sqrt{3}\pi\sqrt{\frac{s}{n}},$$

and it follows that

$$\|[\widehat{\mathbf{b}} - \mathbf{b}^*]_S\|_2 \leq 4\sqrt{3}\pi\sqrt{\frac{s}{n}}$$

holds with probability at least $1 - 1/s$. $\qquad \square$

## D    Additional Lemmas

**Lemma D.1.** Assume that $X_1, \ldots, X_n$ are i.i.d. random vectors following $TE_d(\boldsymbol{\Sigma}^*, \xi; f_1, f_2, \ldots, f_d)$. $\mathbf{T}^*$ is the Kendall's tau matrix defined in Section 3.2, and $\widehat{\mathbf{T}}$ is the Kendall's tau estimator in (3.1). For fixed $\mathbf{x}$ with $\|\mathbf{u}\|_2 \leq 1$, for any $t \leq nC/2$, we have

$$\mathbb{E}\left[\exp\left(t \cdot \mathbf{u}^\top(\widehat{\mathbf{T}} - \mathbf{T}^*)\mathbf{u}\right)\right] \leq \exp\left(\frac{8Ct^2}{n}\right).$$

*Proof.* Denote $S_n$ the group of permutations of $[n]$. For a fixed $\mathbf{u} \in \mathbb{R}^d$ and a permutation $\sigma \in S_n$, define

$$Z_{\sigma,i} = \mathbf{u}^\top\left(\text{sign}\left((\boldsymbol{X}_{\sigma(i)} - \boldsymbol{X}_{\sigma_{i+n/2}})(\boldsymbol{X}_{\sigma(i)} - \boldsymbol{X}_{\sigma_{i+n/2}})^\top\right) - \mathbf{T}^*\right)\mathbf{u}, \quad i = 1, \ldots, n.$$

Observe that

$$\mathbf{u}^\top(\widehat{\mathbf{T}} - \mathbf{T}^*)\mathbf{u} = \frac{1}{n!}\sum_{\sigma \in S_n} \frac{2}{n} \sum_{i \in [n/2]} Z_{\sigma,i}, \tag{D.1}$$

and that for any fixed $\sigma \in S_n$, the $Z_{\sigma,i}$'s are i.i.d. distributed for $i = 1, \ldots, n/2$. We denote the identical distribution as

$$\begin{aligned}
\widetilde{Z} &= \mathbf{u}^\top\big(\operatorname{sign}\big((\boldsymbol{X}_i - \boldsymbol{X}_{i+n/2})(\boldsymbol{X}_i - \boldsymbol{X}_{i+n/2})^\top\big) - \mathbf{T}^*\big)\mathbf{u} \\
&= \big(\mathbf{u}^\top \operatorname{sign}(\boldsymbol{X}_i - \boldsymbol{X}_{i+n/2})\big)^2 - \mathbb{E}\big(\mathbf{u}^\top \operatorname{sign}(\boldsymbol{X}_i - \boldsymbol{X}_{i+n/2})\big)^2.
\end{aligned}$$

Note that $|\mathbf{u}^\top \operatorname{sign}(\boldsymbol{X}_{ik} - \boldsymbol{X}_{i+n/2,k})| \leq \|\mathbf{u}\|_2 \leq 1$ for any $k = 1, \ldots, d$. So $\mathbf{u}^\top \operatorname{sign}(\boldsymbol{X}_{ik} - \boldsymbol{X}_{i+n/2,k})$ is sub-Gaussian and $\widetilde{Z}$ is a centered sub-exponential random variable with $\|\widetilde{Z}\|_{\psi_1} \leq 2$. By Bernstein-type inequality [30], we have

$$\mathbb{E}\exp(t\widetilde{Z}) \leq e^{4Ct^2}, \tag{D.2}$$

where $C$ is an absolute constant. It immediately implies that

$$\begin{aligned}
\mathbb{E}\big[\exp\big(t \cdot \mathbf{u}^\top(\widehat{\mathbf{T}} - \mathbf{T}^*)\mathbf{u}\big)\big] &= \mathbb{E}\bigg[\exp\bigg(\frac{1}{n!}\sum_{\sigma \in S_n} \frac{2t}{n}\sum_{i \in [n/2]} Z_{\sigma,i}\bigg)\bigg] \\
&\leq \frac{1}{n!}\sum_{\sigma \in S_n} \mathbb{E}\bigg[\exp\bigg(\frac{2t}{n}\sum_{i \in [n/2]} Z_{\sigma,i}\bigg)\bigg],
\end{aligned}$$

where the inequality is due to Jensen's inequality. Since for any fixed $\sigma \in S_n$, $Z_{\sigma,i}$'s are i.i.d. distributed and equal to $\widetilde{Z}$ in distribution. We have

$$\begin{aligned}
\mathbb{E}\big[\exp\big(t \cdot \mathbf{u}^\top(\widehat{\mathbf{T}} - \mathbf{T}^*)\mathbf{u}\big)\big] &\leq \bigg(\mathbb{E}\bigg[\exp\bigg(\frac{2t}{n}\widetilde{Z}\bigg)\bigg]\bigg)^{n/2} \\
&\leq \exp\bigg(\frac{8Ct^2}{n}\bigg),
\end{aligned}$$

where the second inequality is due to (D.2). $\qquad\square$

The following lemma is about covering numbers of the sphere.

**Lemma D.2.** [30] The unit Euclidean sphere $S^{n-1}$ equipped with the Euclidean metric satisfies for every $\epsilon > 0$ that

$$|\mathcal{N}_\epsilon| \leq \bigg(1 + \frac{2}{\epsilon}\bigg)^n,$$

where $\mathcal{N}_\epsilon$ is the $\epsilon$-net of $S^{n-1}$.

The following lemma is about how to compute the spectral norm on a $\epsilon$-net.

**Lemma D.3.** [30] Let $\mathbf{A}$ be a symmetric $n \times n$ matrix. For some $\epsilon \in [0, 1/2)$, let $\mathcal{N}_\epsilon$ be an $\epsilon$-net of the unit sphere $S^{n-1}$ in $\mathbb{R}^n$. Then

$$\|\mathbf{A}\|_2 = \sup_{\mathbf{x} \in S^{n-1}} |\mathbf{x}^\top \mathbf{A}\mathbf{x}| \leq (1 - 2\epsilon)^{-1} \sup_{\mathbf{x} \in \mathcal{N}_\epsilon} |\mathbf{x}^\top \mathbf{A}\mathbf{x}|.$$

**Lemma D.4.** [33, 2] There exist vectors $\mathbf{x}_1, \mathbf{x}_2, \cdots$ and $\mathbf{y}_1, \mathbf{y}_2, \cdots$ with $\|\mathbf{x}_i\|_\infty, \|\mathbf{y}_i\|_\infty \leq 1$ for all $i \geq 1$ and a non-negative sequence $a_1, a_2, \cdots$ with $\sum_{i=1}^\infty a_i = 4$, such that $\cos(\pi/2\mathbf{T}) = \sum_{i=1}^\infty a_i\mathbf{x}_i\mathbf{y}_i^\top$.

# E   Additional Experimental Results

In this section, we present the simulation results of Gaussian differential graph model, which is a special case of the semiparametric differential graph models. Note that in the Gaussian case, the

Table 3: Comparisons of estimation errors in Frobenius norm $\|\widehat{\boldsymbol{\Delta}} - \boldsymbol{\Delta}^*\|_F$ for Gaussian differential graph models. N/A means the algorithm did not output the solution in one day.

| Methods | $n = 100, d = 100$ | | $n = 200, d = 400$ | |
| | Setting 1 | Setting 2 | Setting 1 | Setting 2 |
| --- | --- | --- | --- | --- |
| SepGlasso | 33.0574±0.4551 | 56.8891± 0.1778 | 70.1670±0.4316 | 84.9336±0.0025 |
| DPM | 23.5676±0.7222 | 39.4366±0.3814 | N/A | N/A |
| LDGM-L1 | 14.0990±0.6233 | 32.1872±0.4237 | 29.1737±0.4597 | 44.4980±0.5482 |
| LDGM-MCP | 12.4052± 0.5758 | 28.7305± 0.3477 | 27.8458±0.5843 | 38.7960±0.3976 |

Table 4: Comparisons of estimation errors in infinity norm $\|\widehat{\boldsymbol{\Delta}} - \boldsymbol{\Delta}^*\|_{\infty,\infty}$ for Gaussian differential graph models. N/A means the algorithm did not output the solution in one day.

| Methods | $n = 100, d = 100$ | | $n = 200, d = 400$ | |
| | Setting 1 | Setting 2 | Setting 1 | Setting 2 |
| --- | --- | --- | --- | --- |
| SepGlasso | 3.8932±0.1362 | 5.1321±0.0102 | 4.1205±0.1081 | 3.8786±0.0369 |
| DPM | 3.1945±0.0291 | 4.4132±0.1060 | N/A | N/A |
| LDGM-L1 | 2.7127±0.0364 | 4.1265±0.3595 | 2.2423±0.1490 | 3.0224±0.1088 |
| LDGM-MCP | 2.6549± 0.1648 | 3.5277±0.0609 | 2.0638±0.0388 | 2.3904±0.1831 |

inputs for all the methods are the sample covariance matrices $\widehat{\boldsymbol{\Sigma}}_X$ and $\widehat{\boldsymbol{\Sigma}}_Y$ instead of the Kendall's tau based correlation matrices. The ROC curves by averaging the results over 10 repetitions for Gaussian differential graph models are shown in Figure 4, from which we can see our estimator (LDGM-MCP) outperforms the others in all settings. In addition, LDGM-L1 as a special case of our estimator also performs better than DPM and SepGlasso. SepGlasso's performace is poor since it highly depends on the sparsity of both individual graphs. When $n > 100$, the DPM method failed to output the solution in one day. The average results over 10 replicates for estimation in terms of Frobenius norm and infinity norm are summarized in Tables 3 and 4 respectively. Our estimator again achieves smaller error than the other baselines in all settings. In addition, LDGM-L1 also performs better than DPM and SepGlasso.

(a) Setting 1: n=100,d=100 (b) Setting 2: n=100,d=100 (c) Setting 1: n=200,d=400 (d) Setting 2: n=200,d=400

Figure 4: ROC curves for Gaussian differential graph models of all the 4 methods. There are two settings of graph structure. Note that DPM is not scalable to $d = 400$.