[Reviews · NeurIPS 2016]

Reviewer 1

Summary

The present paper deals with differential networking analysis. The authors propose a new model in this framework called "Latent Differential Graph Model" allowing more general networks structures (transelliptical distribution) than just Gaussian. The assumption made throughout the paper, which is now classical in this framework, is the sparsity of the matrix, says $\Delta$, obtained by the difference between the precision matrices of the two networks. This assumption is honest since it means that the network does not change a lot from one state to the other. Here the goal is to estimate $\Delta$ and the authors use some kind of penalized (quasi) log likelihood function minimization. There are two novelties with this criterion: - First, in the cost function the true covariance matrices are estimated by Kendall's tau statistic (other choices are possible and there is no real benefit of using this estimate rather than using the classical sample covariance matrix). - Second, the penalty is non-convex and defined in a general way (see Assumption 4.3). Hence, SCAD and MCP are special case of the procedure. While the penalty is non-convex, the global criterion to be minimized is convex using some constraints on modulus of convexity of the loss function. This makes optimization possible in high dimension. Under classical assumptions, the authors provide upper bounds on the estimation error of $\Delta$ for the infinity and the Frobenius norms. Moreover, numerical studies are performed showing the good behavior of the method.

Qualitative Assessment

An interesting point is that the new model (LDGM) allows more general distribution for networks while adding only small technical difficulties (as compared to Gaussian distributions). I have some comments; 1) Even if the global criterion is convex, it could be interesting to consider the conmputation complexity of the method as compared to critetion with convex penalty (as the $\ell_1$ penalty). 2) Theorems 4.4 and 4.7 are valid for all the penalties satisfying Assumptions 4.3. Since the $\ell_1$ penalty satisfies this assumption, the results are also valid for it. However, the authors argue that the result in Theorem 4.7 is better than recent (a state-of-the-art) results because of the non-convexity of the penalty. This argument fails with the $\ell_1$ penalty. Can the authors makes clearer this point? I think that the result is just "better" because of proofs arguments. Moreover, I believe that the results are not really better since there are two parts in the bound (in Theorem 4.7) and the sparsity parameters $s_1$ and $s_2$ are unknown. 3) Finally, the authors argue quite often that their results and conditions are optimal. Can they provide a reference for lower bounds? For me the factor $M$ appearing in the bounds is tricky. It can be seen as some constant times $s$. Then, it should be taken in consideration in the rates. This is not the case in the present paper. Lastly here is a minor comment: there could be an error in the legend of the second box of Figure 1. As setted, it is the LDGM-L1 the best.

Confidence in this Review

2-Confident (read it all; understood it all reasonably well)


Reviewer 2

Summary

The authors present a new approach to comparing pairs of graph structures. In contrast to previously developed approaches, the proposed approach 1) estimates the difference-graph directly and in a sparse way, thus enabling a more sparse representation of the difference graph (as opposed to estimation of the two graphs first then comparing), 2) the proposed approach relaxes the Gaussian assumption typically made by other methods by using transelliptical distributions, and 3) it uses a family of non-convex penalty functions that allow efficient inference.

Qualitative Assessment

The article is well organized and the results are very thoroughly presented (over 10 pages of proofs). - In the introduction, the authors state estimation of two graphs first, then comparison to estimate the difference, requires estimating twice as many parameters (and therefore twice as many observations) to ensure estimation accuracy. Is this true for the class of methods that regularize the difference between two graphs (e.g. penalize differences in edge weights between two graphs)? - I find Assumption 4.2 to be quite strong and unrealistic – can we expect that the number of non-zero entries in the difference-matrix to not depend on the dimensionality of the data (d)? E.g. Would we expect the number of non-zero entries between graphs of dimension 100 to be on the same order as differences between graphs of 1,000,000 dimensions? - It would be nice if the authors provide some intuition for the significance of Assumption 4.3. - It was unclear to me how sparse the matrices are that were generated from simulations in section 5.1. Also, what is the motivation behind the choice of functions f (line 230)? - The simulations in 5.1 need to explore performance with respect to different sparsity patterns of \Sigma_x, \Sigma_y and delta, since the proposed method is geared towards sparse estimation of the difference-graph. Also, larger graphs (e.g. 1,000 nodes, 10,000 nodes) would be useful as well (since the proposed approach is more likely to be used on high dimensional data). - I found the real-world data application of 5.2 unconvincing for the following reason. The pathway TGFB was hand-chosen, likely because of its extremely high importance in cancer. However, first the validation involved verifying genes involved in differential edges are important in the same kind of cancer. TGF-B is heavily implicated in many, many cancers, and so you would expect many of the genes in that pathway a priori to be involved in cancer to begin with (without even looking at the graph). Second, the validation is of the genes (the nodes), not of the differential edges, which is what the method is inferring. Third, it is well known that hubs in a network tend to be more important (and also more likely to be co-opted by cancer). Hubs in either graph (or both) are also more likely to have a differential edge detected by the method (you can’t have a differential edge adjacent to a node if that node does not have any edge in either graph). So, the method is more likely to pull down a hub node by chance anyways (e.g. a competitive method that just identifies nodes based on degrees in either network may be just as good at pulling down relevant genes). Fourth, there is no mention of false-negative rate (how many ovarian cancer genes were missed by the method/did not have a differential edge). Better validation would have been to identify some genes that are unique to the subtypes studied and looked at their recovery. - Figure 3 was also unconvincing for the same reason. - Real world data is noisy and messy. It is important to record how the networks change for different bootstraps of the samples, or at least the variance for the 5 different 5-fold cross-validations at the optimal values of the tuning parameters of \lambda and b. (The optimal values of \lambda and b should be stated as well.) - The interpretation of the differential networks is unclear, as the edges between genes in the differential networks do not match direct interactions in the KEGG pathways and their relationships are not discussed in the article. Is the motivation simply to find individual genes which are differentially expressed? It is also unclear why the gene CSF2RB found by SepGlasso (p. 8, line 277-9) is “wrong” rather than just “different”. (Searching the NCBI entry for this gene gives a reference to an article on colon cancer.) - Has the data been normalized (i.e. assumed to be Gaussian), or modeled using a transelliptical distribution (as discussed on p.1, line 21-24)? Are the assumptions on the data satisfied, or if not (d > n?), why is it justifiable to use the method on this dataset? - The results for LDGM-MCP seem to recapitulate those from the DPM reference [32]. However, only one gene (e.g. THBS2 rather than COMP and THBS2 for the TGF-beta pathway) appear in the results for DPM in the manuscript, whereas both appear in [32]. Why is there a discrepancy? - In searching the KEGG Apoptosis pathway from (http://www.genome.jp/dbget-bin/www_bget?hsa04210), I could not find gene PRKAR2B or its Entrez gene ID 5577 on the page. - Given that in the ovarian cancer dataset both n and d are small (approx 100), the two precision matrices and their difference could be calculated in this case, for comparison with LDGM-MCP. (The only advantage to estimating the differential network in the manuscript is said to be the factor of 2 in the number of parameters to be estimated, p.2 lines 43-44 and lines 73-74). However, I was wondering if there also an improvement in stability when only the difference is calculated? - I have not seen the Kendall tau statistic used estimations of sample covariance of application papers involving gene expression profiles. Its use seems necessary in the proofs (Lemma C.1), and this should be stated if it is indeed the case. - It would be very useful to provide an implementation of the method (or even better, the source code) to ensure transparency and reproducibility of the results. - There are some minor typos and errors in language/flow of writing. Some of them are listed below. “S = supp(\Delta^*)” appears on p.4, line 122 but “supp” is only defined on p.4, line 141. The definition could be included in the “Notation” section at the end of the Introduction, which was useful. “theories” should be replaced with “theorems” (e.g. p.4 line 141, p.5, line 174) p.4, line 156, “||\Delta||^*_{0,0}” should be replaced by “||\Delta^*||_{0,0}” p. 6, line 227: “by the huge package” (or “by the huge R package” to specify that it is in R), and the version used should also be mentioned, for reproducibility p. 6, line 232, “mdoels” to be replaced by “models”

Confidence in this Review

2-Confident (read it all; understood it all reasonably well)


Reviewer 3

Summary

This paper presents and analyzes a method for estimating the difference network of closely related graphs, corresponding to the difference of two precision matrices. This is done using a generalization of the gaussian the transelliptical distribution. The estimator is based on a non-convex regularized quasi-loglikelihood. It is shown that under specific assumptions this non-convex objective becomes convex allowing for the statistical rate to be analyzed. It is shown that in this regime a fast rate of convergence is achieved.

Qualitative Assessment

The problem of graph difference estimation is interesting and very recently quite popular. Indeed many reasonable regularizers in this setting lead to non-convex penalties. This paper rigorously shows a powerful penalty and that it achieves the oracle properties under mild conditions. I would be interested to obtain more intuition on when the assumptions 4.3 hold. The synthetic and real experiments demonstrate that this method works well compared to natural baselines. The lasso nodewise regression methods of (meinhausen 2006) has been shown to perform better than graphical lasso for estimation of the edge structure in some cases. It would be interesting to see this compared as well if possible. One thing that can be made more clear in the experiments is how the parameters of the estimator are chosen in practice, as this is critical for the guarantees of the method to hold. Overall this is a strong paper that fills an important Gap. There is also some recent literature on this in this years ICML and on arxiv. I don't expect the authors to have addressed these but it would be interesting to consider for a final draft. Fazayeli Generalized Direct Change Estimation in Ising Model Structure. ICML 2016

Confidence in this Review

2-Confident (read it all; understood it all reasonably well)


Reviewer 4

Summary

In this paper, authors propose a novel method which learns the "latent differential graph" between two graphical models, represented by semiparametric elliptical distributions. The estimator is built upon the "quasi-likelihood" and a non-convex regularizer (such as SCAD or MCP). Comparing to a prior work [32], a faster statistical rate is given and performance is verified via synthetic and real-world datasets.

Qualitative Assessment

Authors studied a problem proposed by a prior work [32]: learning the differential graph between two graphical models and propose a novel (though similar) estimator with a non-convex regularizer. Thanks to the rank base estimator and the non-convex regularizer, Authors' method is more general (semi-parametric) and enjoy a faster statistical rate comparing to [32]. In general, this paper is well-written and easy to read. I also have a few comments: 1. Authors proved rates for two scenarios: Thm 4.4 Large magnitude only and Thm 4.7 Large + Small magnitude. However, it would be definitely interesting to include experiments based on these two scenarios, and perhaps the advantage of using a non-convex regularizer would become clearer. 2. It might be interesting to perform a short empirical study on the computational speed of all methods. 3. Although under a slightly different framework, a recent work (see below) also investigated a similar problem. it would be interesting to see some comparisons between these methods in future works. https://www.aaai.org/ocs/index.php/AAAI/AAAI15/paper/viewFile/9454/9942

Confidence in this Review

3-Expert (read the paper in detail, know the area, quite certain of my opinion)


Reviewer 5

Summary

Paper suggests a method which allows to reconstruct a difference between two graphical models producing two distinct data sets. To achieve this goal the authors proposed a Latent Differential Graph Model, where the two Graphical Models themselves are among the latent variables (summed over). The resulting problem is solved by standard (even though nontrivial) optimization methods.

Qualitative Assessment

It would be helpful to extend experimental part, e.g. by comparing with the methods of Ref.[32]. I am not convinced that your approach is advantageous over a simpler one from [32]. I would also like to see comparison with a straightforward methods - where the two graphical models (for two data sets) are reconstructed independently and then the differential graph is retrieved directly. Finally, does your method extends to the case when I have many independent data sets giving different graphical models and I what to learn "robust" graphical structure, i.e. edges present in all (or significant majority) of the graphical models?

Confidence in this Review

1-Less confident (might not have understood significant parts)


Reviewer 6

Summary

The paper presented a graph estimation approach from an interesting perspective. That is not assuming the sparsity in the graph of a single condition but the sparsity in the differential graph of two conditions. Moreover, using more general transelliptical distributions than common Gaussian distributions and a robust Kendall's tau estimate largely broaden the application of the approach for its robustness. Real data analysis also checked the validity of the proposed approach.

Qualitative Assessment

The comments about the paper are as below. 1. Line 115: is assuming that n_X ( the sample number in the first condition ) equals n_Y ( the sample number in the second condition ) necessary? In practice, sample sizes of two experiments are often different and dropping some samples to keep equal sample sizes would lose part of information. I am wondering how to employ the Kendall tau estimator when n_X does not equal to n_Y. 2. As is often the case, some attributes might not have observations in both two conditions. For example, when the attribute is the gene, this gene is measured in the first condition, but it is not measured in the second condition. How to deal with the commonly encountered missing data problem. 3. This paper only provided the approach to detecting differential graph for two conditions. If multiple conditions exist, pairwisely analyzing the graph would cost quadratic computational time. Is it possible to extent the approach to this kind of situation?

Confidence in this Review

2-Confident (read it all; understood it all reasonably well)